# Factors essential for L,D-transpeptidase-mediated peptidoglycan cross-linking and β-lactam resistance in *Escherichia coli*

Jean-Emmanuel Hugonnet[1,2,3], Dominique Mengin-Lecreulx[4], Alejandro Monton[5], Tanneke den Blaauwen[5], Etienne Carbonnelle[1,2,3], Carole Veckerlé[1,2,3], Yves, V. Brun[6], Michael van Nieuwenhze[6], Christiane Bouchier[7], Kuyek Tu[1,2,3], Louis B Rice[8], Michel Arthur[1,2,3]*

[1]INSERM, UMR_S 1138, Centre de Recherche des Cordeliers, Paris, France; [2]Sorbonne Universités, UPMC Université Paris 06, UMR_S 1138, Centre de Recherche des Cordeliers, Paris, France; [3]Université Paris Descartes, Sorbonne Paris Cité, UMR_S 1138, Centre de Recherche des Cordeliers, Paris, France; [4]Institute for Integrative Biology of the Cell (I2BC), CEA, CNRS, Université Paris-Sud, Université Paris-Saclay, Gif-sur-Yvette, France; [5]Bacterial Cell Biology and Physiology, Swammerdam Institute for Life Sciences, University of Amsterdam, Amsterdam, Netherlands; [6]Indiana University, Indiana, United States; [7]Institut Pasteur, Paris, France; [8]Rhode Island Hospital, Brown University, Providence, United States

**Abstract** The target of β-lactam antibiotics is the D,D-transpeptidase activity of penicillin-binding proteins (PBPs) for synthesis of 4→3 cross-links in the peptidoglycan of bacterial cell walls. Unusual 3→3 cross-links formed by L,D-transpeptidases were first detected in *Escherichia coli* more than four decades ago, however no phenotype has previously been associated with their synthesis. Here we show that production of the L,D-transpeptidase YcbB in combination with elevated synthesis of the (p)ppGpp alarmone by RelA lead to full bypass of the D,D-transpeptidase activity of PBPs and to broad-spectrum β-lactam resistance. Production of YcbB was therefore sufficient to switch the role of (p)ppGpp from antibiotic tolerance to high-level β-lactam resistance. This observation identifies a new mode of peptidoglycan polymerization in *E. coli* that relies on an unexpectedly small number of enzyme activities comprising the glycosyltransferase activity of class A PBP1b and the D,D-carboxypeptidase activity of DacA in addition to the L,D-transpeptidase activity of YcbB.

*For correspondence: michel.arthur@crc.jussieu.fr

**Competing interests:** The authors declare that no competing interests exist.

## Introduction

Peptidoglycan, the major component of bacterial cell walls, is a giant ($10^9$ to $10^{10}$ Da) net-like macromolecule composed of glycan strands cross-linked by short peptides (*Turner et al., 2014*) (*Figure 1*). The glycan strands are polymerized by glycosyltransferases and the cross-links are formed by D,D-transpeptidases. The latter enzymes cleave the D-Ala$^4$-D-Ala$^5$ peptide bond of an acyl donor stem and link the carbonyl of D-Ala$^4$ to the side chain amine of diaminopimelic acid (DAP) of the acceptor stem, thereby generating D-Ala$^4$→DAP$^3$ cross-links (*Figure 1A*) (*Pratt, 2008*). β-lactam antibiotics mimic the D-Ala$^4$-D-Ala$^5$ termination of peptidoglycan precursors and inactivate the D,D-transpeptidases by acting as suicide substrates (*Tipper and Strominger, 1965*). The D,D-transpeptidases belong to a diverse family of proteins, commonly referred to as penicillin-binding proteins (PBPs), that fall into three main classes (*Sauvage et al., 2008*). Class A PBPs (PBP1a, 1b, and 1c) combine

glycosyltransferase and D,D-transpeptidase modules. Class B PBPs (PBP2 and 3) are composed of a nonenzymatic morphogenesis module fused to a D,D-transpeptidase module. Class C PBPs (PBP4, 4b, 5, 6, 6b, 7, and AmpH) are monofunctional enzymes with D,D-carboxypeptidase and endopeptidase activities that hydrolyze the D-Ala$^4$-D-Ala$^5$ bond of pentapeptide stems and the D-Ala$^4$→DAP$^3$ bond of cross-linked peptidoglycan, respectively. Peptidoglycan polymerization involves two complexes (*Figure 1B*), the divisome and the elongasome, responsible for septum formation and lateral cell-wall elongation, respectively (*den Blaauwen et al., 2008*). These complexes include all the biosynthetic and hydrolytic enzymes required for the incorporation of new subunits into the expanding peptidoglycan net, as well as cytoskeletal proteins, MreB and FtsZ, acting as guiding devices.

Unusual cross-links connecting two DAP residues were detected in *E. coli* as early as 1969, however the enzymes responsible for their formation at that time were unknown (*Schwarz et al., 1969*). These DAP$^3$→DAP$^3$ cross-links account for 3% and 10% of the cross-links present in the peptidoglycan extracted from bacteria in the exponential and stationary phases of growth, respectively (*Schwarz et al., 1969*). More recently, we have identified the L,D-transpeptidases (Ldt) responsible for the formation of 3→3 cross-links in various bacterial species and shown that these enzymes are structurally unrelated to PBPs (*Mainardi et al., 2008*). Gene deletion and complementation analyses have indicated that the chromosome of *E. coli* encodes five L,D-transpeptidases with distinct functions. Two paralogues form the DAP$^3$→DAP$^3$ peptidoglycan cross-links (YcbB and YnhG) (*Magnet et al., 2008*), whereas the three remaining paralogues anchor the Braun lipoprotein to peptidoglycan (YbiS, ErfK, and YcfS) (*Magnet et al., 2007*).

Here we show that the L,D-transpeptidase activity of YcbB, but not that of YnhG, is able to replace the D,D-transpeptidase activity of all five class A and B PBPs of *E. coli*, leading to β-lactam resistance. We have also identified the various factors required for bypass of the PBPs by YcbB, which include the enzyme partners of the L,D-transpeptidase for peptidoglycan polymerization and upregulation of the (p)ppGpp alarmone synthesis.

## Results

### YcbB-mediated β-lactam resistance

β-lactams of the penam and cephem classes, such as ampicillin and ceftriaxone, respectively, effectively inactivate D,D-transpeptidases belonging to the PBP family. In contrast, L,D-transpeptidases are slowly acylated by these drugs and the resulting acyl-enzymes are unstable (*Triboulet et al., 2013*). This accounts for L,D-transpeptidase-mediated penam and cephem resistance since slow acylation combined with acyl-enzyme hydrolysis results in partial L,D-transpeptidase inhibition. In this study, resistance to ampicillin and ceftriaxone was used to assess the capacity of L,D-transpeptidases to bypass PBPs in *E. coli*. In order to control the level of production of the L,D-transpeptidase YcbB, the corresponding gene was cloned under the control of the IPTG-inducible *trc* promoter of the vector pTRCKm. The resulting plasmid, pJEH11(*ycbB*), was introduced into strain BW25113Δ4 (8), which does not harbor any of the remaining L,D-transpeptidase genes, *i.e. ynhG, ybiS, erfK*, and *ycfS*. Plasmid pJEH11(*ycbB*) did not confer ampicillin resistance to this host in the absence of IPTG or in media containing low concentrations of this inducer (up to 50 μM). Induction of *ycbB* expression with IPTG concentrations greater than 50 μM prevented bacterial growth, indicating that high-level production of the L,D-transpeptidase YcbB was toxic. Selection for ampicillin resistance (32 μg/ml) in the presence of IPTG (500 μM) yielded mutant M1, which was not inhibited by 500 μM IPTG and displayed IPTG-inducible resistance to ampicillin and ceftriaxone (*Figure 2*). Genetic analyses showed that mutant M1 harbors two mutations.

To identify the first mutation, the plasmid was extracted from mutant M1 and introduced into the parental strain *E. coli* BW25113Δ4. Toxicity associated with *ycbB* induction was not observed and the plasmid did not confer ampicillin resistance. Sequencing revealed a mutation located in the vector-born *lacI* gene resulting in an Arg$^{127}$Leu substitution in the inducer binding site. Thus, the plasmid-borne mutation abolished YcbB toxicity by decreasing the level of *ycbB* transcription in inducing conditions. The toxicity associated with high-level production of YcbB may be linked to the putative membrane anchor of the protein as demonstrated for PBP2 in *E. coli* (*Legaree et al., 2007*).

To identify the second mutation, a cured derivative of M1 was obtained by spontaneous loss of the derivative of pJEH11(*ycbB*) harboring the *lacI* mutation, which was designated pJEH11-1(*ycbB*).

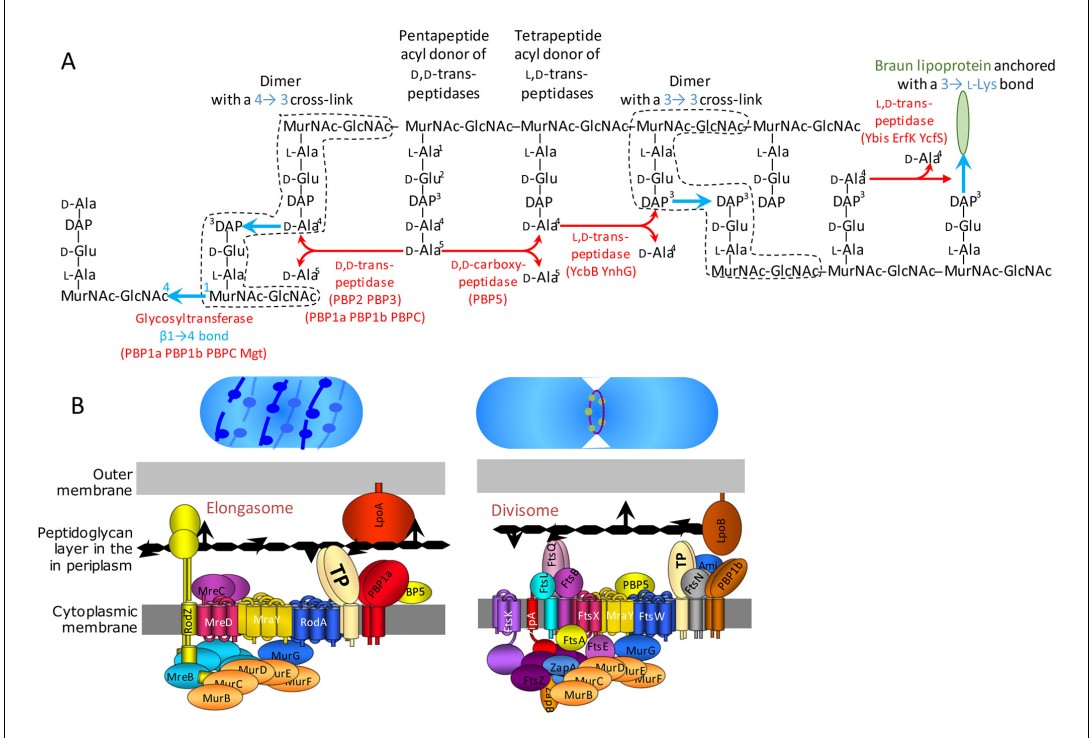

**Figure 1.** Peptidoglycan synthesis in *E. coli.* (A) Reactions catalyzed by enzymes involved in peptidoglycan polymerization and Braun lipoprotein anchoring. (B) Complexes responsible for peptidoglycan synthesis during lateral cell-wall growth and division.

The resulting strain, designated M1$_{cured}$, was susceptible to ampicillin and attempts to select ampicillin-resistant derivatives of that strain were negative (survivor frequency $< 10^{-9}$). Introduction of pJEH11-1(*ycbB*) into M1$_{cured}$ restored ampicillin resistance. These results indicate that a chromosomal mutation is required for ampicillin resistance in addition to the expression of the plasmid copy of *ycbB*.

Testing a large panel of β-lactams using the disk diffusion assay indicated that IPTG-inducible expression of *ycbB*, in combination with the chromosomal mutation, confers broad-spectrum resistance to β-lactams, with the exception of carbapenems (*Table 1*). This conclusion is supported by the phenotypes of the parental strain BW25113Δ4, mutant M1, a derivative of this mutant (M1$_{cured}$) devoid of *ycbB* following the spontaneous loss of plasmid pJEH11-1(*ycbB*), and of a derivative of M1$_{cured}$ expressing *ycbB* following introduction of the plasmid pJEH12(*ycbB*), which is identical to pJEH11-1, except for the origin of replication and the selectable resistance marker.

### Expression of *ynhG* in mutant M1 does not enable emergence of ampicillin resistance

Deletion of both *ynhG* and *ycbB* is required to suppress the in vivo formation of 3→3 cross-links (*Magnet et al., 2008*). Since these results strongly suggest that both genes encode peptidoglycan L,D-transpeptidases, we investigated whether YnhG was able to bypass PBPs, as shown for YcbB. To address this question, the gene *ynhG* was cloned under the control of the *trc* promoter of pTRCKm, and the resulting plasmid was introduced into M1$_{cured}$ and BW25113Δ4. The resulting plasmid did not confer ampicillin resistance in either host. Ampicillin-resistant mutants were not obtained using various concentrations of IPTG and ampicillin (frequency $< 10^{-9}$). These results indicate that bypass of the D,D-transpeptidase activity of the PBPs was only possible with YcbB.

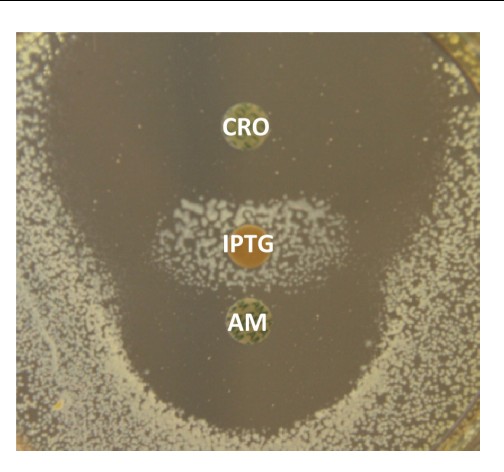

**Figure 2.** IPTG-inducible expression of β-lactam resistance in mutant M1(pJEH11-1). The diffusion assay was performed with disks containing 30 µg of ampicillin (AM), 30 µg of ceftriaxone (CRO), or 10 µg of IPTG.

## Contribution of YcbB to the formation of peptidoglycan cross-links

The respective contributions of PBPs and YcbB to peptidoglycan polymerization were assessed by determining the relative proportions of 4→3 and 3→3 cross-links in mutant M1 (*Figure 3*). The sequence of the cross-links was determined by tandem mass spectrometry analysis of purified peptidoglycan fragments (*Figure 4* and data not shown). In the presence of ampicillin, the D,D-transpeptidase activity of the PBPs was inhibited, and all cross-links were of the 3→3 type. These results indicate that YcbB is sufficient for peptidoglycan cross-linking in the absence of the D,D-transpeptidase activity of the PBPs.

## L,D-transpeptidase activity of YcbB

A soluble fragment of YcbB, devoid of the putative membrane anchor of the protein, was produced in *E. coli*, purified, and assayed for in vitro formation of peptidoglycan cross-links. Incubation of YcbB with a disaccharide-tetrapeptide prepared from the peptidoglycan of *E. coli* resulted in the formation of a peptidoglycan dimer containing a $DAP^3 \rightarrow DAP^3$ cross-link (*Figures 5* and *6A*). Purified YcbB also displayed L,D-carboxypeptidase activity as the enzyme removed the C-terminal residue (D-Ala$^4$) from the tetrapeptide stem (*Figures 5* and *6A*). These results confirm that YcbB is a *bona fide* L,D-transpeptidase, which directly accounts for the synthesis of $DAP^3 \rightarrow DAP^3$ cross-links in mutant M1.

## Inactivation of YcbB by β-lactams

Incubation of YcbB with representatives of the β-lactams belonging to the carbapenem class, *i.e.* meropenem and imipenem, led to full and irreversible acylation of the protein, as detected by mass spectrometry (*Figures 6B* and *7*). In contrast, adducts resulting from acylation of YcbB by β-lactams of the penam and cephem classes were prone to hydrolysis (data no shown), as previously described for Ldt$_{fm}$ from *E. faecium* (*Triboulet et al., 2013*). Thus, the inhibition profile of purified YcbB accounts for broad-spectrum resistance to all β-lactams except carbapenems (*Table 1*).

## Identification of the class C PBP partner of YcbB

Purified YcbB used as the acyl donor a disaccharide-tetrapeptide ending in D-Ala$^4$, but not a disaccharide-pentapeptide ending in D-Ala$^4$-D-Ala$^5$ (*Figure 6A*). We therefore investigated low-molecular-weight class C PBPs to identify the D,D-carboxypeptidase that generates the substrate of YcbB by cleavage of the D-Ala$^4$-D-Ala$^5$ peptide bond of pentapeptide stems. We reasoned that selection for YcbB-mediated ampicillin resistance could be used to determine whether a host strain produces the D,D-carboxypeptidase partner of YcbB for peptidoglycan polymerization. Note that partner refers in the context of this study to an enzyme that is essential for peptidoglycan polymerization in

**Table 1.** Susceptibility of *E. coli* strains determined by the disk diffusion assay.

| Antibiotic | Load (µg) | BW25113 | BW25113Δ4 | M1 IPTG 50 µM | M1<sub>cured</sub> | M1<sub>cured</sub> pJEH12(*ycbB*) | M1<sub>cured</sub> pJEH12(*ycbB*) IPTG 50 µM |
|---|---|---|---|---|---|---|---|
| Amoxicillin | 25 | 23 | 24 | 13 | 28 | 28 | 16 |
| Ampicillin | 10 | 21 | 21 | ND§ | 27 | 26 | ND |
| Amox+Clav† | 20+10 | 20 | 23 | 13 | 27 | 30 | 11 |
| Piperacillin | 75 | 29 | 28 | ND | 36 | 36 | ND |
| Pip+Tazo‡ | 75+10 | 29 | 30 | ND | 37 | 37 | ND |
| Ticarcillin | 75 | 26 | 27 | ND | 37 | 31 | ND |
| Mecillinam | 10 | 17 | 22 | ND | ND | ND | ND |
| Aztreonam | 30 | 33 | 36 | ND | 49 | 47 | ND |
| Cefalotin | 30 | 16 | 18 | ND | 23 | 23 | ND |
| Cefoxitin | 30 | 20 | 25 | 29 | 30 | 30 | 30 |
| Cefotetan | 30 | 30 | 31 | 25 | 41 | 41 | 27 |
| Ceftazidime | 30 | 29 | 30 | ND | 39 | 37 | 9 |
| Cefotaxime | 30 | 33 | 36 | ND | 44 | 44 | 9 |
| Cefixime | 10 | 28 | 30 | ND | 37 | 38 | ND |
| Cefpirome | 30 | 31 | 33 | ND | 41 | 41 | 9 |
| Cefoperazone | 30 | 27 | 28 | ND | 41 | 40 | ND |
| Moxalactam | 30 | 31 | 32 | 15 | 43 | 40 | 17 |
| Ceftriaxone | 30 | 33 | 32 | 15 | 42 | 42 | 18 |
| **Carbapenems** | | | | | | | |
| Doripenem | 10 | 31 | 34 | 32 | 38 | 37 | 38 |
| Meropenem | 10 | 30 | 34 | 35 | 40 | 41 | 34 |
| Imipenem | 10 | 26 | 30 | 35 | 25 | 27 | 28 |
| Ertapenem | 10 | 30 | 35 | 37 | 49 | 47 | 37 |

*BW25113Δ4 is a derivative of *E. coli* BW25113 that does not harbor the *ynhG*, *ybiS*, *erfK*, and *ycfS* genes encoding YcbB paralogues. M1 is a β-lactam-resistant mutant of BW25113Δ4 harboring pJEH11-1(*ycbB*). M1<sub>cured</sub> is a derivative of M1 resulting from the spontaneous loss of pJEH11-1(*ycbB*). Plasmid pJEH12(*ycbB*) was obtained by replacing the origin of replication (ColE1) and resistance marker (kanamycin) of pJEH11-1(*ycbB*) by the p15A replication origin and tetracycline resistance marker of plasmid pACY184. The L,D-transpeptidase gene *ycbB* of pJEH11-1 and pJEH12 are expressed under the control of the IPTG-inducible *trc* promoter and regulated by the LacI Arg[127]Leu repressor.

†Combination of amoxicillin (20 µg) and clavulanate (10 µg).

‡Combination of piperacillin (75 µg) and tazobactam (10 µg).

§ND, not detected as the strains grew at the contact of the disk.

conditions where YcbB is the sole functional transpeptidase without implying that a physical contact between YcbB and that protein exists or is essential for enzyme activity. We therefore introduced plasmid pJEH12(*ycbB*) into the mutant from the Keio collection that harbors a kanamycin-resistance cassette in place of *dacA* encoding class C PBP5 (12). Plating the resulting strain on agar containing ceftriaxone (32 µg/ml) and IPTG (50 µM) led to no survivors (frequency <10$^{-9}$), unless a plasmid copy of *dacA* was provided *in trans* (9 × 10$^{-6}$) (*Table 2*). Thus, the D,D-carboxypeptidase PBP5 was essential for YcbB-mediated β-lactam resistance. In contrast, independent deletions of *dacB*, *dacC*, and *dacD* encoding PBP4, PBP6, and PBP6b, respectively, had no impact on the frequency of selection of ceftriaxone-resistant mutants (*Table 2*). Further analyses were performed with strain CS801-4 (13), which was obtained by deletions of *dacA*, *dacB*, *dacC*, *dacD*, *mrcA*, *ampH*, and *ampC* (*Table 3*). In this host, plasmids pJEH12(*ycbB*) and pTrc99AΩ*dacA* were both required for selection of

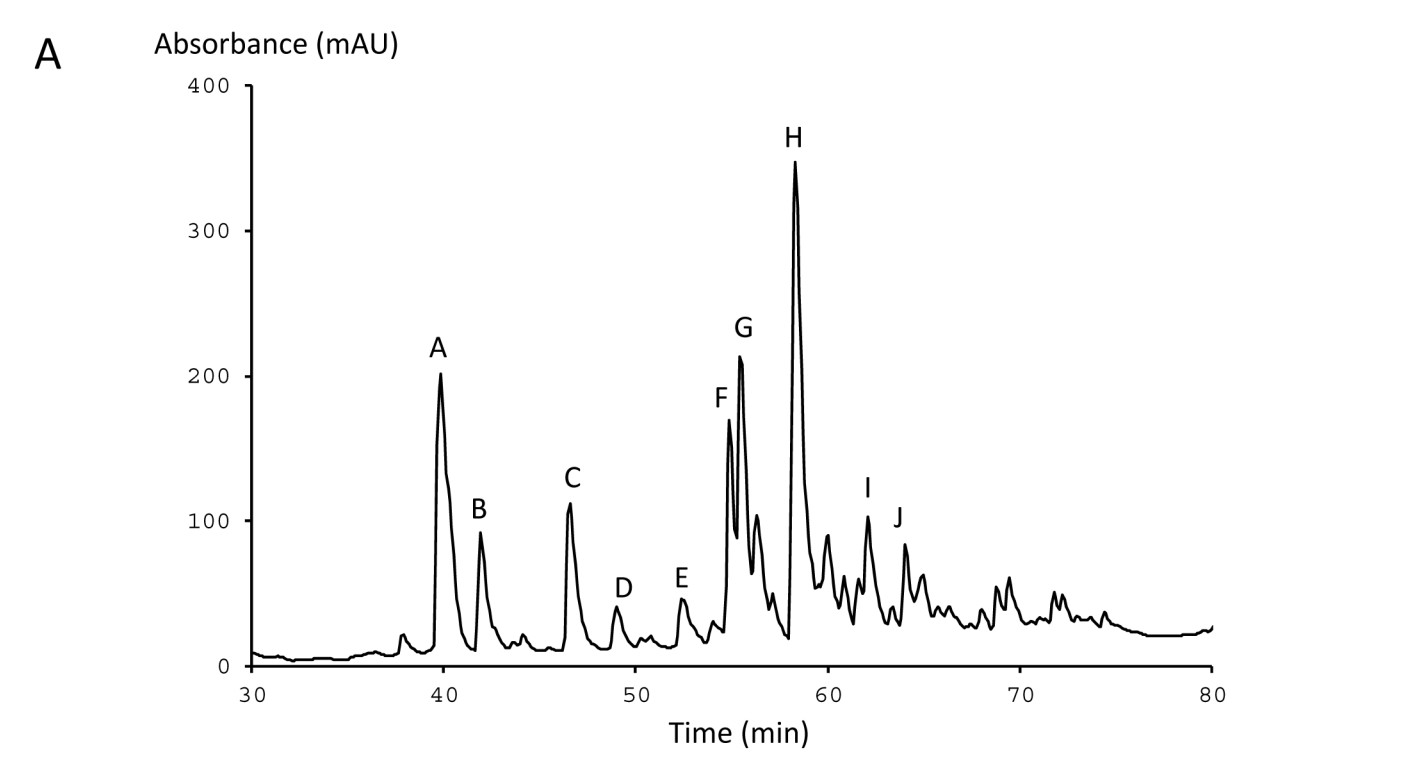

**Figure 3.** Peptidoglycan composition of mutant M1 grown in presence of ampicillin (16 μg/ml). (**A**) *rp*HPLC profile of disaccharide-peptides. Absorbance was recorded at 210 nm (mAU, absorbance units x 10³). (**B**) Identification of disaccharide-peptides by mass spectrometry. The relative abundance of peptidoglycan fragments was estimated as the percentage of the total integrated area. The observed and calculated monoisotopic mass of muropeptides is indicated in Da. GM, GlcNAc-MurNAc; [anh], anhydro; [R], reduced; Tri, tripeptide L-Ala-γ-D-Glu-DAP (DAP, diaminopimelic acid); Tetra, tetrapeptide L-Ala-γ-D-Glu-DAP-D-Ala; Tri-Gly, tetrapeptide L-Ala-γ-D-Glu-DAP-Gly; Penta, pentapeptide L-Ala-γ-D-Glu-DAP-D-Ala-D-Ala; 3→3, cross-link generated by L,D-transpeptidation.

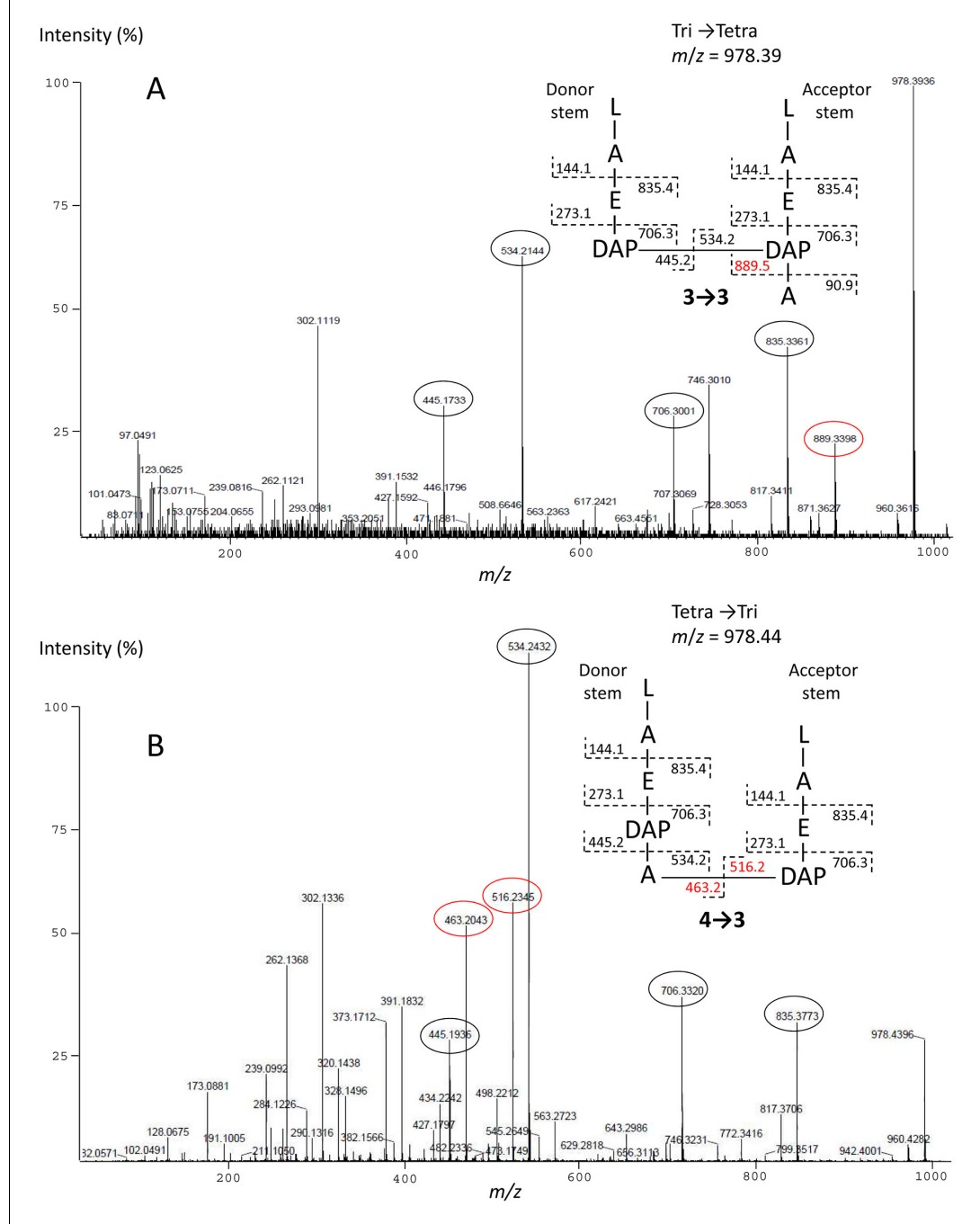

**Figure 4.** Sequencing of the peptidoglycan cross-link of lactoyl-peptides by tandem mass spectrometry. The figure illustrates the capacity of the method to discriminate isomers containing 3→3 (A) and 4→3 (B) cross-links. Fragments specific of each isomer are shown in red. L, D-Lac; A, L-Ala or D-Ala; a, C-terminal D-Ala; E, D-Glu; DAP, diaminopimelic acid. All dimers present in peptidoglycan preparations from mutant M1 were identified by this method.

ceftriaxone-resistant mutants. These results indicate that the D,D-carboxypeptidase activity of PBP5 is necessary and sufficient to generate the essential tetrapeptide-containing substrate of YcbB. It is of note that PBP5 is only inhibited by high concentrations of β-lactams (*Curtis et al., 1979*). This accounts for the proposed participation of PBP5 in peptidoglycan synthesis in the presence of β-

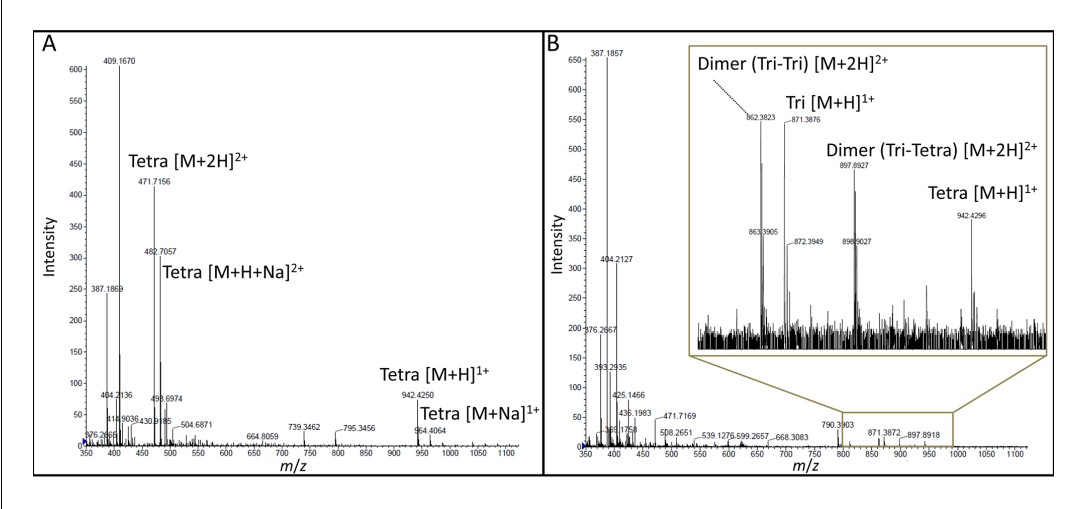

**Figure 5.** Mass spectrometry analysis of the products of the reactions catalyzed in vitro by YcbB. (**A**) Disaccharide-tetrapeptide used as the substrate. The muropeptide GlcNAc-MurNAc-L-Ala[1]-γ-D-Glu[2]-DAP[3]-D-Ala[4] (Tetra) was purified from the *E. coli* cell wall peptidoglycan. The peak at *m/z* 942.4250 [M+H][1+] corresponds to an observed mass ($M_{obs}$) of 941.42 Da in agreement with the calculated mass ($M_{cal}$) of 941.41 Da. (**B**) Reaction products. YcbB (10 μM) was incubated with the disaccharide-tetrapeptide (30 μM) for 2 hr at 37°C revealing the formation of (i) the tripeptide GlcNAc-MurNAc-L-Ala[1]-γ-D-Glu[2]-DAP[3] (Tri; $M_{obs}$ = 870.40 Da; $M_{cal}$ = 870.37 Da) by the L,D-carboxypeptidase activity of YcbB; (ii) the peptidoglycan dimer bis-disaccharide-Tri-Tetra ($M_{obs}$ = 1793.80 Da; $M_{cal}$ = 1793.77 Da) by the L,D-transpeptidase activity of YcbB; and the peptidoglycan dimer bis-disaccharide-Tri-Tri ($M_{obs}$ = 1722.78 Da; $M_{cal}$ = 1722.73 Da) by the L,D-transpeptidase and L,D-carboxypeptidase activities of YcbB.

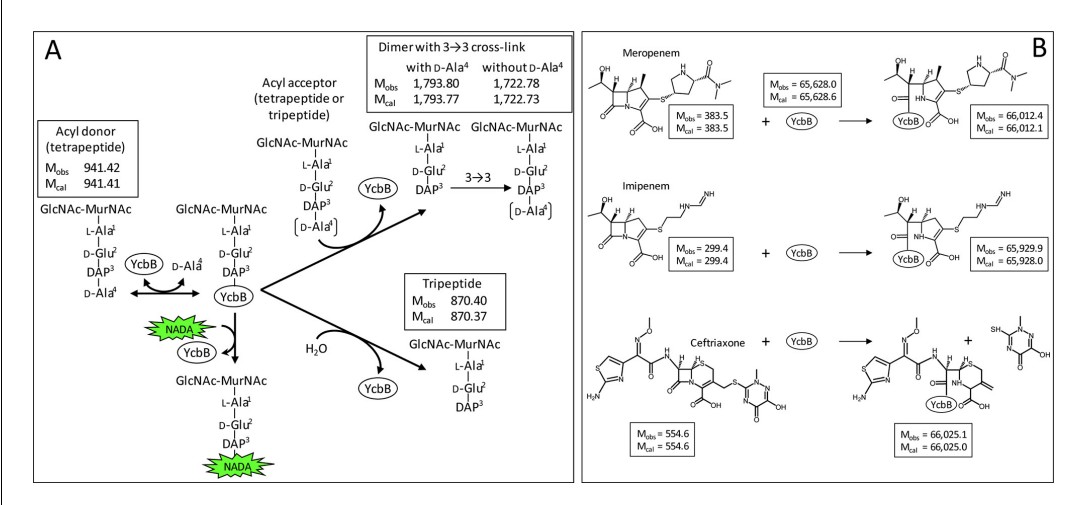

**Figure 6.** Reactions catalyzed by the L,D-transpeptidase, YcbB. (**A**) In vitro cross-linking assay. Incubation of YcbB with the reduced disaccharide GlcNAc-MurNAc-tetrapeptide resulted in the formation of a dimer containing a 3→3 cross-link (L,D-transpeptidase activity). YcbB also hydrolyzed the C-terminal D-Ala[4] residue of tetrapeptide stems (L,D-carboxypeptidase activity). The muropeptides were determined by mass spectrometry. The observed ($M_{obs}$) and calculated ($M_{cal}$) monoisotopic masses are indicated in Daltons. The pentapeptide GlcNAc-MurNAc-L-Ala[1]-γ-D-Glu[2]-DAP[3]-D-Ala[4]-D-Ala[5] was not a substrate of YcbB. The reaction used to label peptidoglycan with a fluorescent derivative of D-Ala (NADA) is indicated. (**B**) Acylation of YcbB by β-lactams. Incubation of YcbB with two carbapenems, *i.e.* meropenem and imipenem, led to the acyl-enzymes shown, which were stable. In contrast, the acyl-enzyme formed with ceftriaxone was unstable, accounting for resistance of mutant M1 to this cephalosporin. The observed ($M_{obs}$) and calculated ($M_{cal}$) average masses are indicated in Daltons. No adduct was observed with amoxicillin.

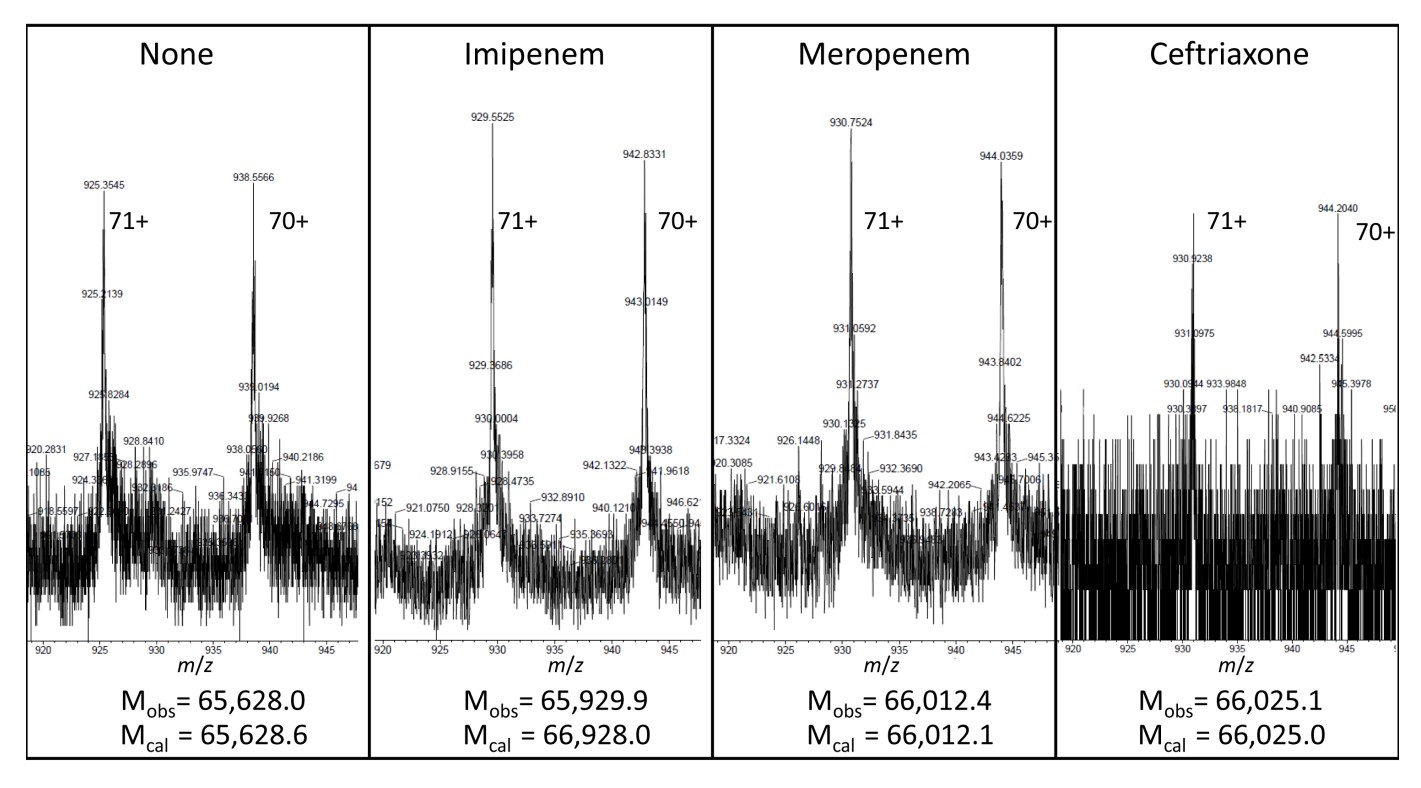

**Figure 7.** Mass spectrometry analyses of the adducts resulting from acylation of YcbB by the β-lactams imipenem, meropenem, and ceftriaxone. YcbB (10 µM) was incubated with β-lactams (100 µM) for 1 hr at 37°C. The peaks correspond to the $[M+71H]^{71+}$ and $[M+72H]^{72+}$ ions. The observed ($M_{obs}$) and calculated ($M_{cal}$) masses are indicated. Acylenzymes formed with ceftriaxone (a cephem) were detected although they were slowly hydrolyzed, as previously described for Ldt$_{fm}$ from *E. faecium* (**Triboulet et al., 2013**). Acylenzymes formed with ampicillin (a penam) were not detected by mass spectrometry.

lactams. This also accounts for the residual activity of cefoxitin (**Table 1**, inhibition zone of 30 mm) since this cephalosporin displays moderate activity against PBP5 (14).

## Identification of the glycosyltransferase partner of YcbB

Since the L,D-transpeptidase YcbB does not harbor a glycosyltransferase-related domain, peptido-glycan polymerization in the presence of ampicillin is predicted to require cooperation between the L,D-transpeptidase activity of YcbB and the glycosyltransferase activity of housekeeping enzymes. *E. coli* produces four candidate enzymes for the latter function, including three class A PBPs (PBP1a, PBP1b, and PBP1c) and a monofunctional glycosyltransferase (MgtA) (**Figure 1A**) encoded by the genes *mrcA*, *mrcB*, *pbpC*, and *mgtA*, respectively (**Sauvage et al., 2008**). To determine which of these four enzymes cooperate with YcbB, plasmid pJEH12(*ycbB*) was introduced into four mutants of the Keio collection harboring a kanamycin-resistance gene cassette in place of each of the four glycosyltransferase genes. Ceftriaxone-resistant mutants were obtained with the *mrcA*, *pbpC*, and *mgtA* mutants, but not with the *mrcB* mutant, indicating that PBP1b is an essential partner of YcbB for peptidoglycan polymerization in the presence of β-lactams.

PBP1b is a bifunctional class A PBP containing transpeptidase and glycosyltransferase modules. To assess the role of the transpeptidase activity of PBP1b, the *mrcB* gene was introduced into plasmid p*Trc*99A, and the transpeptidase module was inactivated by the substitution S$^{510}$A (**Terrak et al., 1999**). It has been previously shown that PBP1b harboring this substitution is able to catalyze in vitro the polymerization of lipid II into glycan strands in the absence of TPase activity (**Terrak et al., 1999**; **Egan et al., 2015**). The derivatives of p*Trc*99A encoding wild-type PBP1b and PBP1b S$^{510}$A were introduced into the strain BW25113Δ*mrcB* harboring pJEH12(*ycbB*). Selection of

**Table 2.** Selection of β-lactam-resistant derivatives of *E. coli* BW25113 harboring various plasmids.

| Deletion[*] | Plasmid 1 | Plasmid 2 | Frequency x 10$^{9}$[†] |
|---|---|---|---|
| None | pACYC184 | pTrc99A | <1 |
| None | pJEH12(*ycbB*) | none | 3200 |
| *dacA* | pJEH12(*ycbB*) | none | <1 |
| *dacA* | pJEH12(*ycbB*) | pTrc99AΩ*dacA* | 9100 |
| *dacB* | pJEH12(*ycbB*) | none | 13,000 |
| *dacC* | pJEH12(*ycbB*) | none | 9000 |
| *dacD* | pJEH12(*ycbB*) | none | 9100 |
| *mrcB* | pJEH12(*ycbB*) | pTrc99A | <1 |
| *mrcB* | pJEH12(*ycbB*) | pTrc99AΩ*mrcB* | 6400 |
| *mrcB* | pJEH12(*ycbB*) | pTrc99AΩ*mrcB* TG (E$^{233}$M) [‡] | <1 |
| *mrcB* | pJEH12(*ycbB*) | pTrc99AΩ*mrcB* TP (S$^{510}$A)[§] | 1700 |
| *mrcA* | pJEH12(*ycbB*) | none | 1600 |
| *pbpC* | pJEH12(*ycbB*) | none | 2700 |
| *mgtA* | pJEH12(*ycbB*) | none | 2700 |
| *lpoA* | pJEH12(*ycbB*) | none | 5000 |
| *lpoB* | pJEH12(*ycbB*) | none | <1 |

[*]Mutants of the Keio collection harboring a kanamycin-resistance gene cassette in place of the indicated gene.
[†]Frequency of survivors obtained by plating 10$^{9}$ colony forming units on agar containing ceftriaxone (32 µg/ml) and IPTG (50 µM). Values are the median from 2 to 8 experiments.
[‡]The glycosyltransferase (GT) module of PBP1b was selectively inactivated by the E$^{233}$M amino acid substitution.
[§]The transpeptidase (TP) module of PBP1b was selectively inactivated by the S$^{510}$A substitution.

ceftriaxone-resistant mutants was obtained in both cases. Thus, the glycosyltransferase activity of PBP1b, but not its D,D-transpeptidase activity, is required for YcbB-mediated β-lactam resistance. We also tested PBP1b harboring the E$^{233}$M substitution inactivating the glycosyltransferase domain of the protein using the same approach. As expected, ceftriaxone-resistant mutants were not obtained with derivative of BW25113Δ*mrcB* producing YcbB and PBP1b E$^{233}$M. The limitation of the latter observation is that no or significantly reduced transpeptidase activity is observed when the glycosyltransferase of *E. coli* class A PBPs is not functional, although the latter enzymes still bind β-lactam antibiotics indicating a properly folded transpeptidase domain (*Terrak et al., 1999*; *Egan et al., 2015*; *Born et al., 2006*; *Bertsche et al., 2005*; *den Blaauwen et al., 1990*). PBP1b E$^{233}$M may therefore be deficient in both transpeptidase and transglycosylase activities.

**Table 3.** Selection of β-lactam-resistant derivatives of *E. coli* CS801-4 harboring various plasmids.

| Plasmid 1 | Plasmid 2 | Frequency x 10$^{9}$ |
|---|---|---|
| pACYC184 | pTrc99A | <1 |
| pJEH12(*ycbB*) | None | <1 |
| pJEH12(*ycbB*) | pTrc99A | <1 |
| none | pTrc99AΩ*dacA* | <1 |
| pJEH12(*ycbB*) | pTrc99AΩ*dacA* | 5900 |

*E. coli* CS801-4 harbors deletions of genes *pbp4, 5, 6, 7, mrcA, ampH, ampC*, and *dacD*

To determine whether PBP1b was sufficient for resistance, we constructed by serial deletions a derivative of BW25113 lacking chromosomal copies of *mrcA*, *mrcB*, *pbpC*, and *mgtA*. This mutant, designated BW25113Δ4GT/pMAK705Ω*mrcB*, was viable at 30°C owing to the expression of a plasmid copy of *mrcB* harbored by the vector pMAK705, which is thermosensitive for replication. Mutants displaying IPTG-inducible ceftriaxone resistance were obtained by plating BW25113Δ4GT harboring pMAK705Ω*mrcB* and pJEH12(*ycbB*) at 30°C on agar containing ceftriaxone (32 µg/ml) and IPTG (50 µM) (frequency of survivors of $5 \times 10^{-7}$). Mutants were also obtained at 37°C if pMAK705Ω*mrcB* was replaced by p*Trc*99AΩ*mrcB*, which is not thermosensitive for replication. Together, these results indicate that the glycosyltransferase activity of PBP1b is necessary and sufficient for glycan-chain elongation when YcbB is the only functional cross-linking enzyme.

The outer membrane lipoproteins LpoA and LpoB have been shown to be essential for the activity of class A PBP1a and PBP1b, respectively (*Paradis-Bleau et al., 2010*; *Typas et al., 2010*). As expected, ampicillin-resistant mutants were only obtained with the *lpoA* mutant (*Table 2*), indicating that LpoB is essential for the participation of PBP1b in glycan-chain elongation in the context of peptidoglycan cross-linking by YcbB.

## Bypass of the D,D-transpeptidase activity of class B PBPs by YcbB

*E. coli* produces two class B PBPs, PBP2 and PBP3, which are essential and specifically inhibited by mecillinam (*Spratt, 1977*; *Vinella et al., 1993*) and aztreonam (*Georgopapadakou et al., 1982*), respectively. In the presence of IPTG, mutant M1 was resistant to mecillinam and aztreonam, indicating that inhibition of the D,D-transpeptidase module of the class B PBPs did not impair YcbB-mediated ampicillin resistance (*Table 1* and data not shown). Surprisingly, M1 was resistant to mecillinam in the absence of IPTG. M1$_{cured}$ was also resistant to mecillinam, indicating that resistance to this antibiotic was mediated by a chromosomal mutation independently from YcbB production (*Table 1*). This observation raised the possibility that the same mutation might be responsible both for mecillinam resistance in the absence of YcbB in M1$_{cured}$ and for broad-spectrum resistance to penams and cephems in M1 following the induction of the plasmid copy of *ycbB* by IPTG. To test this hypothesis, derivatives of strain BW25113/pJEH12(*ycbB*) were selected on media containing mecillinam alone (50 µg/ml) or ampicillin (32 µg/ml) and IPTG (50 µM). Independent mutants obtained on each selective medium were analyzed for expression of β-lactam resistance. Four of the seven mutants selected on mecillinam remained susceptible to ampicillin both in the presence or absence of IPTG. The three remaining mecillinam-resistant mutants (M2, M3, and M4) displayed IPTG-inducible ampicillin resistance, as did mutant M1. All six mutants selected on ampicillin-and-IPTG-containing media (M5 to M10) were resistant to mecillinam in the absence of IPTG and, additionally, resistant to ampicillin in its presence (as was M1). These results indicate that acquisition of a mutation conferring mecillinam resistance is necessary and sufficient for YcbB-mediated ampicillin resistance. In contrast, only a portion of the mutations conferring mecillinam resistance enabled YcbB-mediated ampicillin resistance.

**Table 4.** Mutations detected in M1 to M7.

| Mutant | Selection | Position | Mutation | Impact |
|---|---|---|---|---|
| M1 | Ap | 22,373 | Δ13-nt[*] | IleRS translation |
| M2 | Me | 1,960,069 | T→C | I$^3$T in ArgRS |
| M3 | Me | 1,801,022 | A→C | S$^{517}$A in ThrRS |
| M4 | Ap | 2,520,555 | G→T | R$^{40}$S in GluRS |
| M5 | Ap | 2,520,540 | C→T | D$^{45}$N in GluRS |
| M6 | Ap | 1,948,853 | G→T | T$^{557}$N in AspRS |
| M7 | Ap | 1,950,015 | G→A | P$^{170}$S in AspRS |

[*]13-base pair deletion (positions 22,373 to 22,385).

Ap, ampicillin; IleRS, isoleucine-tRNA synthetase; Me, mecillinam.

## Identification of chromosomal mutations essential for YcbB-mediated ampicillin resistance

Whole genome sequencing revealed that mutants M1 to M7 each harbored a single mutation (*Table 4*). This observation confirms that a single mutation is responsible for mecillinam resistance in the absence of YcbB and for YcbB-mediated broad-spectrum β-lactam resistance upon *ycbB* induction by IPTG. Mutant M1 harbored a 13-bp deletion located in an untranslated region upstream from the isoleucine-tRNA synthetase (IleRS) gene. The remaining mutants (M2 to M7) harbored missense mutations in genes encoding ArgRS, ThrRS, GluRS, and AspRS, with two distinct substitutions for the latter two enzymes.

The deletion present in M1 most probably impairs translation of IleRS since it ends 5 nucleotides upstream from the ATG codon and includes the ribosome binding site. Thus, the mutation present in M1 impaired aminoacylation of tRNA$^{Ile}$, a defect known to result in activation of (p)ppGpp synthesis by RelA in response to ribosomal stalling, following occupancy of the ribosome acceptor site by uncharged tRNAs (*Hauryliuk et al., 2015*). The same conclusion is also likely to apply to missense mutations in the aminoacyl-tRNA synthetase genes of mutants M2 to M7, although we cannot exclude the possibility that the corresponding amino acid substitutions altered other properties of the proteins.

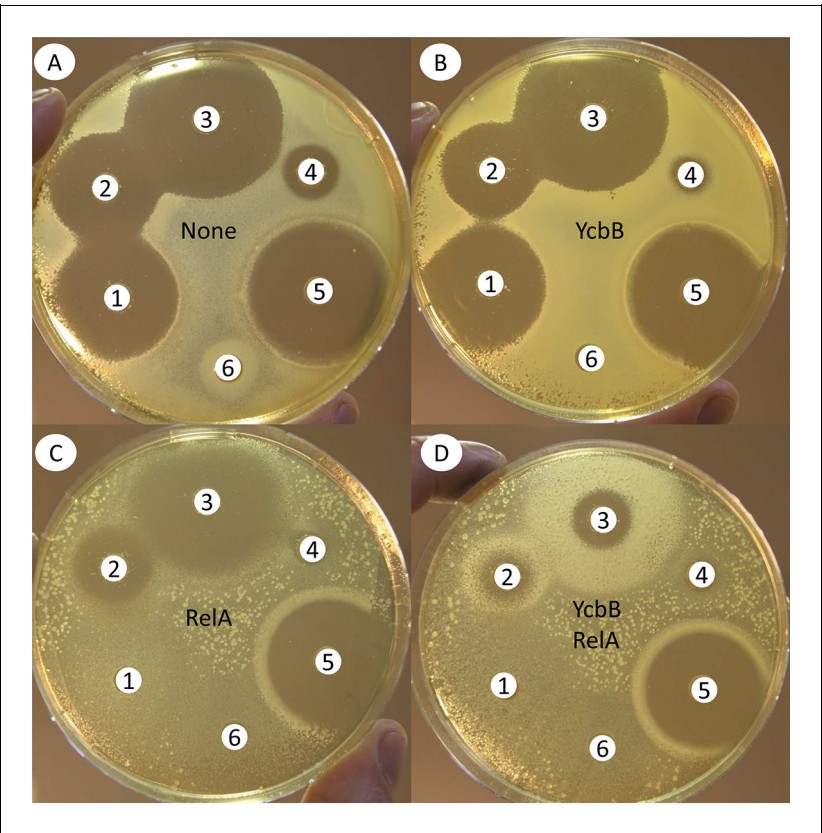

**Figure 8.** Impact of induction of RelA 1–455 and YcbB synthesis on the activity of β-lactams. Antibiograms using the disk diffusion assay were performed on BHI agar (**A**) supplemented with 50 μM IPTG (**B**), 1% arabinose (**C**) or both inducers (**D**), to induce expression of *ycbB* and *relA* 1–455 encoded by compatible plasmids pKT2 and pKT8, respectively. Disks were loaded with 10 μg of mecillinam (1), 10 μg of ampicillin (2), 30 μg of ceftriaxone (3), 30 μg of tetracycline (4), 10 μg of imipenem (5), or 30 μg of chloramphenicol (6). Plates were inoculated with BW25113Δ*relA* harboring plasmids pKT8(*relA*) and pKT2(*ycbB*).

## Production of (p)ppGpp triggers expression of YcbB-mediated ampicillin resistance

In order to modulate the level of synthesis of (p)ppGpp, a fragment of *relA* encoding the first 455 residues of the protein was cloned under the control of the arabinose-inducible promoter of vector pBAD33. The truncated protein encoded by the resulting plasmid, pBAD33Ω*relA* 1–455, is devoid of the C-terminal ribosome binding module and constitutively synthesizes (p)ppGpp (*Schreiber et al., 1991*). The latter plasmid and a compatible plasmid coding for IPTG-inducible synthesis of YcbB, pKT2(*ycbB*), were introduced into a derivative of *E. coli*, BW25113, lacking the chromosomal copy of *relA*. In the absence of both inducers, the resulting strain was susceptible to β-lactams, as was the parental strain BW25113 (*Figure 8A*). Induction of YcbB synthesis by IPTG had no impact on the activity of β-lactams (*Figure 8B*). Induction of RelA 1–455 synthesis by arabinose led to expression of resistance to mecillinam, but not to other β-lactams (*Figure 8C*). Induction of both RelA 1–455 and YcbB recapitulated the phenotype of mutant M1 (*Figure 8D*). These results indicate that YcbB production and increased (p)ppGpp synthesis are both required for bypass of the D,D-transpeptidase activity of PBPs and broad-spectrum β-lactam resistance. Production of (p)ppGpp by RelA 1–455 recapitulates the phenotype conferred by mutations in the aminoacyl-tRNA synthetase genes of mutants M1 to M7, in agreement with their proposed role in the activation of (p)ppGpp synthesis by RelA in response to binding of uncharged tRNAs to the ribosome acceptor site.

## Localization of YcbB activity

L,D-transpeptidases catalyze the exchange of D-Ala[4] at the C-terminus of tetrapeptide stems by various D-amino acids (*Mainardi et al., 2005*). The first step in this reaction involves nucleophilic attack of the carbonyl of DAP[3] by the catalytic Cys of the enzyme, leading to the release of D-Ala[4] and formation of a thioester bond (*Figure 6A*). In the second step, the resulting acyl-enzyme is attacked by the D-amino acid, leading to the formation of a DAP-D-amino acid peptide bond. We used a fluorescent derivative of D-Ala (NADA) (*Kuru et al., 2012*) to localize YcbB activity in live bacteria. As shown in *Figure 9*, intense labeling was detected at mid-cell indicating that YcbB and its tetrapeptide substrate are co-localized in the septum. As controls, we also showed that labeling was inhibited by meropenem, which inactivates YcbB and PBPs, but not by ampicillin, which only inactivates PBPs.

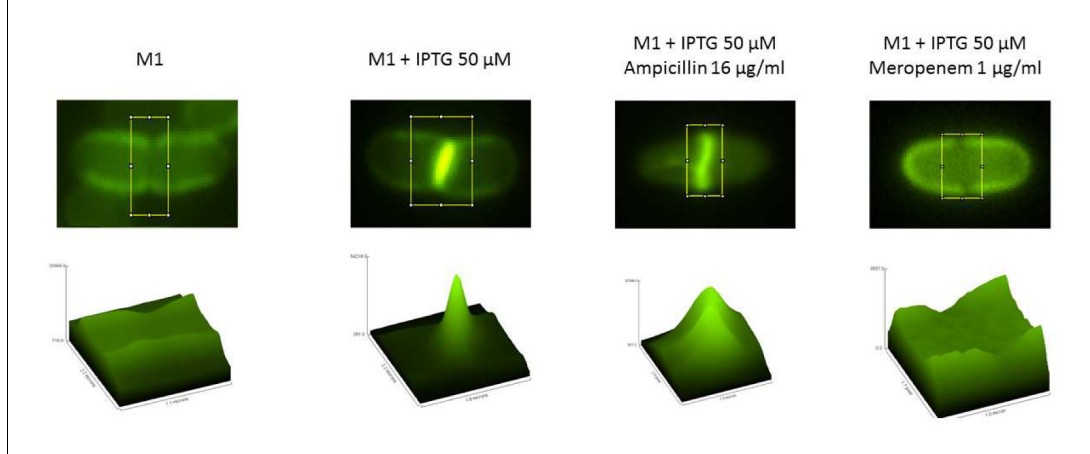

**Figure 9.** Localization of YcbB activity based on labeling of peptidoglycan with a fluorescent derivative of D-Ala (NADA). Mutant M1 was grown in the absence or in the presence of 50 µM IPTG to induce *ycbB* expression. Prior to labeling with NADA, bacteria were incubated with ampicillin, which inhibits PBPs but not YcbB, or meropenem, which inhibits all transpeptidases. The graphics in the lower panel correspond to the surface plot within the yellow rectangle.

## Discussion

*E. coli* is known to resist β-lactams through a combination of drug inactivation by β-lactamases and reduced access of the drugs to their targets following activation of efflux pumps and modification of outer membrane porins. Here we show that *E. coli* can also resist β-lactams through bypass of the D, D-transpeptidase activity of high-molecular-weight PBPs. The resistance mechanism required production of the L,D-transpeptidase YcbB, which was purified and shown to catalyze the formation of peptidoglycan cross-links in vitro (*Figure 6A*) and to be inactivated only by carbapenems (*Figure 6B*), accounting for broad-spectrum resistance to other classes of β-lactams (*Table 1*). The bypass mechanism involves substantial modification of the set of enzymes required for peptidoglycan polymerization. In wild-type strains, lateral expansion of the peptidoglycan network and the formation of the septum involve two protein complexes, the elongasome and the divisome, respectively, which integrate distinct sets of peptidoglycan polymerases with partially overlapping functions (*den Blaauwen et al., 2008*) (*Figure 1*). In contrast, the L,D-transpeptidase activity of YcbB is sufficient for peptidoglycan cross-linking in mutant M1. This conclusion is based on the exclusive detection of DAP$^3$→DAP$^3$ cross-links in the peptidoglycan of M1 grown in the presence of ampicillin (*Figures 3* and *4*) and on mutagenic or chemical invalidation of the transpeptidase module

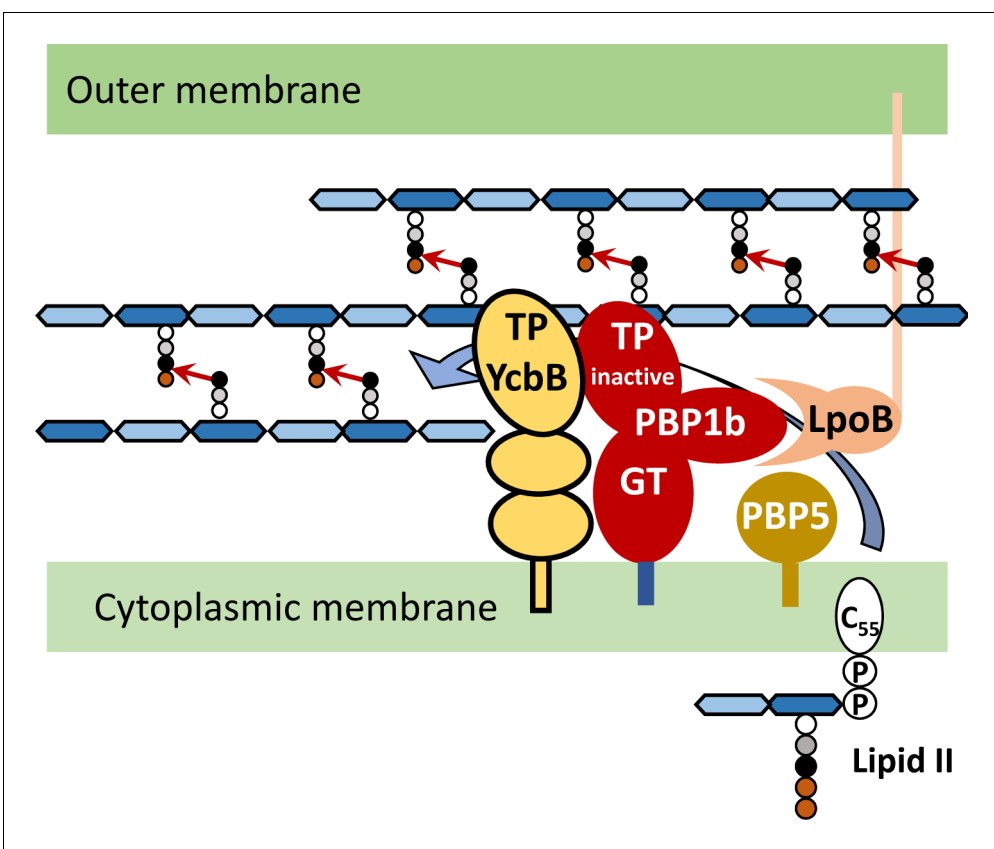

**Figure 10.** Peptidoglycan polymerization in mutant M1. The lipid intermediate II (Lipid II) consists in the disaccharide-pentapeptide subunit linked to the undecaprenyl lipid transporter (C$_{55}$) by a pyrophosphate bond. *N*-acetylglucosamine linked to *N*-acetylmuramic acid (MurNAc) by a β1→4 bond is represented by light and dark blue hexagons, respectively. The pentapeptide stem is linked to the D-lactoyl group of MurNAc and assembled by the sequential addition of L-Ala (white circle), D-Glu (grey circle), *meso*-diamopimelic acid (DAP; black circle), and the dipeptide D-Ala-D-Ala (orange circles). Following translocation through the cytoplasmic membrane, the subunit is polymerized by the glycosyltransferase (GT) activity of PBP1b, which requires binding of LpoB to its UBH2 domain (*Vinella et al., 1993*). The transpeptidase (TP) activity of PBP1b and other PBPs is bypassed by the TP activity of YcbB that forms 3→3 cross-links (arrows) connecting DAP residues at the 3rd position of stem peptides. The donor substrate of YcbB is generated by the D,D-carboxypeptidase activity of PBP5.

of class-A or -B PBPs (*Table 2* and *Table 1*). In wild-type strains, glycan chains are polymerized by either PBP1a or PBP1b, although the former is preferentially associated with the elongasome and the latter with the divisome (*den Blaauwen et al., 2008*). In contrast, PBP1b, together with its cognate lipoprotein LpoB, are both essential and sufficient for peptidoglycan polymerization in mutant M1 exposed to ampicillin. Peptidoglycan labeling with a fluorescent D-amino acid indicated that YcbB activity is preferentially located in the septum, as is its PBP1b partner (*Figure 9*). The cross-linking and glycosyltransferase activities of YcbB and PBP1b were nonetheless sufficient for lateral extension of the peptidoglycan network since bacterial cells retained a rod shape in the presence of ampicillin. This observation indicates that the cross-linking activity of high-molecular-weight PBPs is not indispensable to the maintenance of cell shape in *E. coli*. Together, our data identified a minimum set of enzyme activities that are necessary and sufficient for peptidoglycan polymerization in the presence of β-lactams (*Figure 10*). This set comprises the L,D-transpeptidase activity of YcbB, the glycosyltransferase activity of PBP1b, and the D,D-carboxypeptidase activity of PBP5, which provides the essential tetrapeptide substrate of YcbB. Bypass of the D,D-transpeptidase activity of PBPs by YcbB is potentially useful to explore the function of members of the polymerization complexes. For example, the fact that LpoB remains essential in the L,D-transpeptidation context indicates that it directly stimulates the glycosyltransferase activity of PBP1b, as indicated in a recent study (*Markovski et al., 2016*).

In addition to YcbB production, increased synthesis of the alarmone (p)ppGpp by RelA was required for broad-spectrum β-lactam resistance. This conclusion was drawn from the construction of strains that conditionally produce YcbB and a truncated form of RelA (*Figure 8*) (*Table 5*) and by an independent selection of mutants with impaired aminoacyl-tRNA synthetases (*Table 4*). Prior analyses also showed that PBP2 is not indispensable in *E. coli* when (p)ppGpp synthesis is increased, leading to mecillinam resistance (*Vinella et al., 1992*). Aside from mecillinam resistance, increased (p)ppGpp synthesis has also been shown to regulate growth rate and several different stress responses, including the response to exposure to antibiotics (*Hauryliuk et al., 2015*). The latter leads to persistence, a response that involves production of rare cells within a bacterial population that remain susceptible but are not killed by the drugs, in the absence of acquisition of any mutation. In *E. coli*, tolerance to ampicillin mediated by increased (p)ppGpp synthesis leads to a 1000-fold reduction in bacterial killing. The bacterial cells are tolerant to numerous antibiotics, including non-β-lactams, such as ciprofloxacin, which acts on DNA topoisomerases (*Maisonneuve et al., 2013*). The cascade triggered by elevated (p)ppGpp involves an increase in the intracellular concentration

**Table 5.** Minimal inhibitory concentration of β-lactams against *E. coli* strains harboring various plasmids[*].

| β-lactam | Inducer[†] | Strains | | | | | | | |
|---|---|---|---|---|---|---|---|---|---|
| | | BW25113 | BW25113 pJEH12(*ycbB*) | M1cured | M1cured pJEH12(*ycbB*) | BW25113 pKT8(*relA'*) | BW25113 pKT8(*relA'*) pKT2(*ycbB*) | BW25113ΔrelA pKT8(*relA'*) | BW25113ΔrelA pKT8(*relA'*) pKT2(*ycbB*) |
| Ampicillin | None | 8 | 8 | 8 | 8 | 8 | 4 | 8 | 4 |
| | IPTG | 8 | 8 | 8 | 128 | 8 | 4 | 8 | 4 |
| | Ara | 8 | 8 | 8 | 8 | 8 | 8 | 8 | 4 |
| | IPTG+Ara | 8 | 8 | 8 | 128 | 8 | 64 | 8 | 128 |
| Ceftriaxone | None | 0,05 | 0,05 | 0,05 | 0,05 | 0,05 | 0,05 | 0,05 | 0,05 |
| | IPTG | 0,05 | 0,05 | 0,05 | 32 | 0,05 | 0,05 | 0,05 | 0,05 |
| | Ara | 0,05 | 0,05 | 0,05 | 0,05 | 0,05 | 0,1 | 0,05 | 0,05 |
| | IPTG+Ara | 0,05 | 0,05 | 0,05 | 32 | 0,05 | 32 | 0,05 | 32 |

[*]Minimal inhibitory concentrations were determined by the agar dilution method with an inoculum of $10^5$ colony forming units per spot in brain heart infusion agar after 24 hr of incubation at 37°C. The same results were obtained with $10^4$ and $10^6$ colony forming units, indicating that YcbB in combination with elevated (p)ppGpp rendered the bulk of the population resistant.

[†]Induction was performed with 50 mM IPTG, 1% arabinose (Ara), and a combination of both inducers (IPTG+Ara). IPTG induces expression of the *ycbB* gene of plasmids pJEH12(*ycbB*) and pKT2(*ycbB*). Arabinose induces expression of *relA'* encoding the first 455 residues of RelA.

of polyphosphate via inhibition of the exopolyphosphatase PPX. In turn, elevated polyphosphate stimulates the protease activity of Lon, which degrades antitoxins, thereby activating the toxins that prevent bacterial killing through inhibition of translation and bacterial growth. This cascade is distinct from YcbB-mediated ampicillin resistance in several respects. Persistence emerges from stochastic events that render a limited number of bacterial cells transiently tolerant to multiple families of antibiotics, whereas YcbB in combination with elevated (p)ppGpp rendered the bulk of the population resistant only to β-lactams that did not inactivate the L,D-transpeptidase activity of YcbB (*Table 5*). In addition, plasmids pKT2(*ycbB*) and pKT8(*relA* 1–455) conferred ampicillin and ceftriaxone resistance to mutants from the Keio collection deficient in production of the Lon protease and the polyphosphate kinase PPK (data not shown) indicating that tolerance and resistance do not involve the same (p)ppGpp-regulated circuits.

## Materials and methods

### Strains, plasmids, and growth conditions

All strains were derived from *E. coli* BW25113 (32), except strain CS801-4 (13), kindly provided by Kevin D. Young. Single-deletion strains were obtained from the Keio collection (*Baba et al., 2006*), provided by the National BioRessource Project-*E.coli* at National Institute of Genetics (Shigen, Japan). Serial deletion of class A *pbp* genes was obtained by the one-step technique described by B. Wanner and K. Datsenko (*Datsenko and Wanner, 2000*) using the primers depicted in *Table 6.* Strains were grown in brain heart infusion broth or agar at 37°C unless otherwise specified.

**Table 6.** Oligonucleotides used in this study.

| Oligonucleotide | Sequence |
| --- | --- |
| PBP1b-inact1 | GAAGAACAGAAAATCGGGCTTTTGCGCCTGAATATTGCGGAGAAAAAGCCATATGAATATCCTCCTTAG |
| PBP1b-inact2 | GTTATTTTACCGGATGGCAACTCGCCATCCGGTATTTCACGCTTAGATGGTGTAGGCTGGAGCTGCTTC |
| PBP1a-inact1 | GCGCGTTTGTTTATAAACTGCCCAAATGAAACTAAATGGGAAATTTCCACATATGAATATCCTCCTTAG |
| PBP1a-inact2 | CAAGTGCACTTTGTCAGCAAACTGAAAAGGCGCCGAAGCGCCTTTTTAAGTGTAGGCTGGAGCTGCTTC |
| PBPC-inact1 | ATGCCTCGCTTGTTAACCAAACGCGGCTGCTGGATAACGTTGGCAGCCGCATATGAATATCCTCCTTAG |
| PBPC-inact2 | CCGTCAAATGCAGGGTCACGTTGCGCCCGCGTTCAGTTAACGGTTCGCCGTGTAGGCTGGAGCTGCTTC |
| MGT-inact1 | GCGGCATTGATAAGCTGGTTTCCCGCGTGCTGGTTCTGGCTGAATGAGTCATATGAATATCCTCCTTAG |
| MGT-inact2 | TCGTGAGAGCAAAACGCTGGCCCTCACTTCGCGCGAAGCTTAATCCAGCGTGTAGGCTGGAGCTGCTTC |
| PBP1b-NcoI | AAAACCATGGCCGGGAATGACCGCGAGCCAATTGGACGC |
| PBP1b-SacI | GTTATGAGCTCGGATGGCAACTCGCCATCCGGTATTTCACGC |
| PBP1a-BspHI | TTCTCATGAAGTTCGTAAAGTATTTTTTGATCCTTGC |
| PBP1a-SacI | AGCCGGAGCTCGCGTTCACGCCGTATCCGGCATAAACAAGTGCAC |
| PMAK-PBP1b | CCAAGGATCCGTAAGGTTGGTTTTCTCCCTCTCCCTGTGGG |
| PBP1b-TG1 | GCGACCATGGACCGTCATTTTTACGAGCATGATGGAATC |
| PBP1b-TG2 | CGGTCCATGGTCGCCAGCAAAGTATCCACCAGCAAATCC |
| PBP1b-TP1 | TCGATTGGCGCCCTTGCAAAACCAGCGACTTATCTGACGGC |
| PBP1b-TP2 | TGCAAGGGCGCCAATCGAACGACGCGCCTGCATCGCACGG |
| RelA-XbaI | AATCTAGAAATCGATGGTACTTTTCTC |
| RelA-PstI | AACTGCAGCTACAGCTGGTAGGTGAACGGC |
| DacA-SacI | GGGAGCTCGGCATCTGATGTGTCAAT |
| DacA-BamHI | GGGGATCCTTAACCAAACCAGTGATGG |
| CDF-for | AACTGCAGTCATGAGCGGATACATAT |
| CDF-rev | AAATCGATATCTAGAGCGGTTCAGTAG |

## Construction of plasmids expressing the L,D-transpeptidase gene *ycbB*

To avoid incompatibilities between antibiotic resistance markers and replication origins of the various plasmids used in this study, three plasmids encoding YcbB were constructed.

Construction of pJEH11(*ycbB*) was based on vector pTRCKm, a derivative of *pTrc*99A obtained by replacing the ampicillin-resistance gene by the kanamycin-resistance gene of pCR-Blunt (Invitrogen, Carlsbad, CA) (our laboratory collection). The gene *ycbB* of BW25113 carried by plasmid p*Trc*99AΩ*ycbB* (Turner et al., 2014) was subcloned into the vector pTRCKm using restriction endonucleases NcoI and XbaI. The resulting plasmid was designated pJEH11(*ycbB*). A derivative of pJEH11(*ycbB*), designated pJEH11-1(*ycbB*), was obtained upon selection on p*Trc*99AΩ*ycbB* containing ampicillin and IPTG. Plasmid pJEH11-1(*ycbB*) carries a mutation in the *lacI* gene resulting in an Arg[127]Leu substitution in the inducer binding site of the LacI repressor.

To construct pJEH12(*ycbB*), the ApaI-XmnI fragment of pJEH11-1(*ycbB*) harboring *lacI* Arg[127]Leu, *ycbB*, and their respective promoters were cloned between the EcoRI and ScaI sites of pACYC184. For this construction, the EcoRI and ApaI protruding ends were blunted by treatment with the Klenow fragment of *E. coli* DNA polymerase I. The resulting plasmid, pJEH12(*ycbB*), contains the p15A replication origin, a tetracycline resistance marker, *lacI* Arg[127]Leu, and *ycbB* expressed under the control of the IPTG-inducible *trc* promoter.

To construct pKT2(*ycbB*), the origin of replication CDF was amplified by PCR with oligonucleotides CDF-for and CDF-rev (Table 6) from vector pCDFDuet-1 (Novagen). PstI and ClaI restriction sites were included at the 5′ extremity of the forward and reverse primers, respectively. To replace the p15A replication origin of pJEH12(*ycbB*), the plasmid was digested with ClaI and PstI, and ligated to the CDF amplified PCR product digested with PstI and ClaI. The resulting plasmid, pKT2 (*ycbB*), contains the CDF replication origin, a tetracycline resistance marker, *lacI* Arg[127]Leu, and *ycbB* expressed under the control of the *trc* promoter.

## Peptidoglycan structure

Peptidoglycan was extracted by the boiling SDS method and digested with muramidases (Magnet et al., 2008). The resulting muropeptides were reduced, separated by *rp*HPLC, collected, lyophilized, and identified by electrospray time-of flight mass spectrometry (Q-star Pulsar, Applied Biosystem) (Magnet et al., 2008). For tandem mass spectrometry, muropeptides were treated with ammonia to generate lactoyl-peptides following cleavage of the ether bond internal to *N*-acetylmuramic acid (MurNAc) (Magnet et al., 2008). Lactoyl-peptides were separated by *rp*HPLC, collected, lyophilized, and identified by mass spectrometry. The sequence of the cross-links was determined by tandem mass spectrometry using $N_2$ as the collision gaz (collision of 36–40 eV) (Magnet et al., 2008).

## Purification of the L,D-transpeptidase, YcbB

The *ycbB* gene was amplified by PCR and cloned into pETMM82 (33) between the EcoRI and XhoI restriction sites. The fusion protein comprises the disulfide isomerase DsbC, a 6×His tag, a TEV protease cleavage site, and residues 30-616 of YcbB. The enzyme was produced in *E. coli* BL21(DE3) following induction by IPTG (0.5 mM) at 16°C for 19 hr. The fusion protein was purified from clarified lysates by nickel affinity chromatography in 25 mM Tris-HCl, 300 mM NaCl, pH 8.0 (buffer A). DsbC and the 6×His tag were removed by TEV protease digestion followed by nickel affinity chromatography. YcbB was further purified as a monomer by size-exclusion chromatography (Superdex 75 HiLoad 26/60, GE Healthcare) in buffer A.

## Mass spectrometry analyses of YcbB acylation by β-lactams

The formation of drug-enzyme adducts was tested by incubating YcbB (10 µM) with β-lactams (100 µM) at 20°C in 100 mM sodium-phosphate buffer (pH 6.0) for 1 hr. Mass spectra were acquired in the positive mode (Qstar Pulsar I, Applied Biosystem) as previously described (Triboulet et al., 2013).

## In vitro peptidoglycan cross-linking assay

The disaccharide-tetrapeptide GlcNAc-MurNAc-L-Ala[1]-γ-D-Glu[2]-DAP[3]-D-Ala[4] was purified from the *E. coli* BW25113 cell-wall peptidoglycan (30 µM) and incubated with YcbB (10 µM) for 2 hr at 37°C in

25 mM Tris-HCl, 300 mM NaCl (pH 8.0). The reaction mixture was desalted (ZipTip $C_{18}$; Millipore) and analyzed by mass spectrometry (*Triboulet et al., 2015*).

## Deletion of glycosyltransferase genes

Inactivation of the genes *mrcA*, *mrcB*, *pbpC*, and *mgt* was performed by the Datsenko and Wanner technique (*Pratt, 2008*) with PCR products obtained by amplification of the kanamycin-resistance cassette with oligos PBP1a-inact1 and PBP1a-inact2 (*mrcA*), PBP1b-inact1 and PBP1b-inact2 (*mrcB*), PBPC-inact1 and PBPC-inact2 (*pbpC*), and MGT-inact1 and MGT-inact2 (*mgt*).

## Construction of p*Trc*99A derivatives expressing the class A *mrcB* PBP gene

Gene *mrcB* was amplified with oligonucleotides PBP1b-NcoI and PBP1b-SacI. The amplicon was digested with SacI and NcoI and cloned into p*Trc*99A digested by the same enzymes.

## Construction of a pMAK705 derivative expressing *mrcB*

pMAK705 is a replication thermosensitive plasmid (rep[ts]) (*Tipper and Strominger, 1965*). *mrcB* was amplified with oligonucleotides PMAK-PBP1b and PBP1b-SacI, and the amplicon was digested with BamHI and SacI, and cloned into pMAK705 digested with the same enzymes.

## Inactivation of the transpeptidase module of PBP1b

A 5' portion of the *mrcB* gene was amplified from plasmid p*Trc*99AΩ*mrcB* with oligonucleotides PBP1b-NcoI and PBP1b-TP2. The primer PBP1b-TP2 introduced a NarI site and replaced the codon specifying the catalytic residue Ser[510] by an Ala codon. This amplicon, designated A1, was digested with NcoI and NarI. The remaining 3' portion of the *mrcB* gene was amplified with primers PBP1b-SacI and PBP1b-TP1, which comprises a NarI site. The amplicon designated A2 was digested with NarI and SacI. The amplicons A1 and A2 were ligated to p*Trc*99A digested with NcoI and SacI.

## Inactivation of the glycosyltransferase module of PBP1b

For inactivation of the glycosyltransferase module of PBP1b, a 5' portion of the *mrcB* gene was amplified from plasmid p*Trc*99AΩ*mrcB* with primers PBP1b-NcoI and PBP1b-TG1. The primer PBP1b-TG1 introduced a NcoI site and replaced the codon specifying the catalytic residue Glu[233] by a Met codon. The amplicon, designated B1 was digested with NcoI and HindIII. The remaining 3' portion of the *mcrB* gene was amplified with primers PBP1b-SacI and PBP1b-TG2, which comprises a HindIII site. The amplicon, designated B2 was digested with HindIII and SacI. The amplicons B1 and B2 were ligated to p*Trc*99A digested with NcoI and SacI.

## Construction of the plasmid pBAD33Ω*relA* 1–455 expressing a truncated form of RelA

A fragment of *relA* encoding residues 1–455 was amplified by PCR (primers RelA-XbaI and RelA-PstI) and cloned into vector pBAD33 using restriction endonucleases XbaI and PstI. The recombinant plasmid encodes a truncated version of RelA (residues 1 to 455), which is unable to bind to the ribosome and retains (p)ppGpp synthase activity (*Sauvage et al., 2008*).

## Selection of β-lactam resistant mutants

*E. coli* strains were grown overnight in 10 ml of BHI broth at 37°C, concentrated by centrifugation, and *ca.* $10^9$ colony-forming units were plated on agar containing ampicillin (32 µg/ml) or ceftriaxone (32 µg/ml) and IPTG (50 µM). Colony-forming units were enumerated after 48 hr incubation at 37°C. To determine the frequency of survivors, cultures were also plated on BHI agar to enumerate bacteria present in the inoculum.

## Assay for the identification of YcbB partners

In order to identify proteins essential for YcbB-mediated peptidoglycan cross-linking in the absence of the D,D-transpeptidase activity of PBPs, we expressed *ycbB* in *E. coli* mutants from the Keio collection, which were obtained by replacement of individual open-reading frames by a kanamycin-resistance gene cassette (*den Blaauwen et al., 2008*). We hypothesized that insertion of the Km

cassette in place of a gene encoding an essential partner of YcbB would produce a mutant unable to acquire ampicillin resistance despite production of the L,D-transpeptidase (frequency $< 10^{-9}$ on agar containing 32 µg/ml ampicillin or 32 µg/ml ceftriaxone, and 50 µM IPTG). For non-essential genes, mutants displaying IPTG-inducible expression of β-lactam resistance were obtained at a frequency $\geq 10^{-6}$).

## Whole genome sequencing

Genomic DNA was prepared with the Wizard Genomic DNA purification kit (Promega). Mutations were identified using the Illumina single reads sequencing technology. Library preparation was performed with Genomic DNA Sample Prep Kit v1. Nebulization was used to share genomic DNA, and fragments were blunt-ended, phosphorylated, and A-tailed prior to ligation of sequencing adapters. Fragments with an insert size of *ca.* 200 bp were gel-extracted and enriched with PCR (14 cycles) before library quantification and validation. Clusters were generated by hybridization of the library to the flow cell and bridge amplification. Single reads of 36 cycles were collected on a GAIIX (Illumina, San Diego, CA). The Illumina Analysis Pipeline (version 1.6) was used for image analysis, base calling, and error estimation. Raw sequence files were filtered using programs developed by N. Joly (Biology IT Center, Institut Pasteur, Paris). Quality-filtered trimmed reads (minimum 50 bases) were mapped on the genome sequence of *E. coli* BW25113 (ID: 2829859), and variant detection was performed with CLC Genomics Workbench version3 (CLC Bio, Denmark). Raw sequencing data have been deposited in the Dryad Digital Repository database (*Bouchier et al., 2015*).

## Probing of YcbB activity in live bacteria with a fluorescent D-amino acid

M1$_{cured}$ harboring pJEH12(*ycbB*), was grown in the absence or presence of 50 µM IPTG in lysogeny broth (LB) to an optical density at 600 nm of 0.3 at 37°C. The bacterial culture was further incubated with 16 µg/ml ampicillin or 1 µg/ml meropenem at 37°C for 30 and 15 min, respectively. A fluorescent derivative of D-Ala containing a 7-nitrobenzofurazan fluorophore (NADA; 1 mM) (*Kuru et al., 2012*) was added to the culture under constant shaking and incubation was continued for 150 s at 37°C. Cells were washed three times in LB (5000 rpm; 5 min), immobilized on 0.1 x LB containing 1% agarose, and incorporation of NADA was determined with a Nikon Eclipse T1 microscope (Nikon Plan Fluor × 100/1.30 Oil Ph3 DLL objective) coupled to an EMCCD camera (Hamamatsu Flash 4.0). Images were further analyzed by ImageJ (NIH).

## Acknowledgements

Mass spectra were collected at the mass spectrum facility of the Muséum National d'Histoire Naturelle. We thank Dr. Matthias Bochtler for the generous gift of plasmid pETMM82. This work was supported by the National Institute of Allergy and Infectious Diseases (Grant RO1 307 AI046626) to LBR and MA, the project NAPCLI from the JPI AMR program (ZonMW project 60-60900-98-207 for AM), and the National Institutes of Health (Grant GM113172) to MSvN. We thank Laurence Ma and Magali Tichit for technical assistance for genome sequencing. The Genomics Platform is a member of the 'France Génomique' consortium supported by the Agence Nationale de la Recherche (ANR10-INBS-09-08).

## Additional information

### Funding

| Funder | Grant reference number | Author |
|---|---|---|
| National Institute of Allergy and Infectious Diseases | RO1 307 AI046626 | Louis B Rice<br>Michel Arthur |
| Joint Program Initiative on Antimicrobial Research | ZonMW project 60-60900-98-207 | Alejandro Monton |
| Joint Program Initiative on Antimicrobial Research | NAPCLI | Jean-Emmanuel Hugonnet<br>Alejandro Monton<br>Tanneke den Blaauwen<br>Michel Arthur |

| National Institutes of Health | GM113172 | Michael van Nieuwenhze |

The funders had no role in study design, data collection and interpretation, or the decision to submit the work for publication.

## Author contributions

J-EH, Conception and design, Acquisition of data, Analysis and interpretation of data, Drafting or revising the article; DM-L, Conception and design, Acquisition of data, Analysis and interpretation of data; AM, Acquisition of data, Analysis and interpretation of data, Drafting or revising the article; TdB, MA, Conception and design, Analysis and interpretation of data, Drafting or revising the article; EC, Substantial contribution for the acquisition of data (construction of mutants, determination of survival frequencies), Author critically read the manuscript and approved the final version, Acquisition of data; CV, Substantial contribution for the acquisition of data (bacterial cultures, peptidoglycan preparation), Author critically read the manuscript and approved the final version, Acquisition of data; Y,VB, MvN, Substantial contribution for essential reagents (fluorescent labelled amino acids), Author critically read the manuscript and approved the final version, Contributed unpublished essential data or reagents; CB, Substantial contribution for genome sequencing and analysis of the sequence data, Author critically read the manuscript and approved the final version, Acquisition of data, Analysis and interpretation of data; KT, Substantial contribution for acquisition of data (mutants constructions, survival frequencies determination, MCIs determination), Author critically read the manuscript and approved the final version, Acquisition of data; LBR, Substantial contribution for analysis of data and interpretation, Author critically read the manuscript and approved the final version, Analysis and interpretation of data

## Author ORCIDs

Michel Arthur, http://orcid.org/0000-0003-1007-636X

## Additional files

### Major datasets

The following dataset was generated:

| Author(s) | Year | Dataset title | Dataset URL | Database, license, and accessibility information |
|---|---|---|---|---|
| Bouchier C, Hugonnet JE, Arthur M | 2015 | Raw genomic sequences of mecillinam and resistant mutants of E. coli | http://dx.doi.org/10.5061/dryad.t5r8m | Available at Dryad Digital Repository under a CC0 Public Domain Dedication |

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
