## [Decision Letter]

Thank you for submitting your article "Factors essential for L,D-transpeptidase mediated peptidoglycan crosslinking and β-lactam resistance in *Escherichia coli*" for consideration by *eLife*. Your article has been favorably evaluated by Wendy Garrett as the Senior Editor and three reviewers, including Jose Lemos (Reviewer #3) and Michael S. Gilmore, who is a member of our Board of Reviewing Editors.

The reviewers have discussed the reviews with one another and the Reviewing Editor has drafted this decision to help you prepare a revised submission.

Summary:

This is a generally well written and technically highly competent manuscript for which there would be a high level of enthusiasm if the issues below can be adequately addressed. It describes the conditions necessary to generate a complete, ampicillin-resistant cell wall by employing a previously identified L,D-transpeptidase activity mediated by YcbB, in conjunction with increased alarmone levels. The importance of this work is that peptidoglycan biosynthesis represents a complex activity that is uniquely microbial, which can be targeted by molecules that do not require translocation across the cytoplasmic membrane. The contribution of YcbB to generation of the unusual 3-3 crosslinks that ordinarily constitutes ~ 10% of the *E. coli* peptidoglycan was discovered by this group in 2008 (11). The present work extends that to show that this activity, combined with an elevation in alarmone ppGpp, PBP1b and DacA expression, are sufficient for complete cell wall biosynthesis in a manner that is resistant to ampicillin inhibition. A main attraction for this theme is the problem of antibiotic resistance, and this pathway renders the cell ampicillin resistant in vitro.

Essential revisions:

A major limitation of the manuscript in its current form is that are some gaps in logic and omissions of critical data. For example, there is broad interest in the field in antibiotic resistance, and the central theme of this paper is the ampicillin-resistant generation of a complete cell wall. However, this reviewer is unfamiliar with that happening in nature, and as a result, most β-lactam related resistance is mediated by drug modifying enzymes and efflux – is that the case? For the sake of a balanced perspective, the authors should discuss any evidence that this bypass occurs, or if not, why not?

In terms of leaps of logic in the experimentation that require further explanation:

1) As it currently reads (subsection “YcbB-mediated β-lactam resistance”, first paragraph), it appears that selection for both resistance to 500 μg/ml IPTG induction and resistance to 32 μg/ml ampicillin were performed simultaneously. Is this correct? If so, the rationale as well as rate of mutation should be better explained and documented, as this is central to the manuscript.

2) Was an actual change in transcription in the *lacI* mutant seen or is this simply presumed? (And why is high expression of YcbB toxic (subsection “YcbB-mediated β-lactam resistance”, first paragraph)? The authors may not know, of course, but it would be useful to briefly speculate based on the authors understanding).

3) Was a change in ppGpp levels in the tRNA synthetase mutants that were essential for the phenotype actually measured? As there are broad effects of altering ppGpp pools, the lack of direct connection detracts from the perceived robustness of the results and the model proposed. There are also effects on the cell of mistranslation.

4) When the glycosyltransferase (GT) activity of PBP1B is blocked by mutation, its transpeptidase (TP) activity also decreases substantially or disappears (Egan et al., 2015). This means that the PBP1B-E^233^ mutant lacks both GT and TP activities. Thus, even if the GT enzymatic domain is "inactivated independently" (subsection “Identification of the glycosyltransferase partner of YcbB”, second paragraph), it is unclear how the authors can claim that only "the GT activity of PBP1b.… is required for.… β-lactam resistance" (second paragraph), or that this GT activity "is necessary and sufficient" (third paragraph) [Reference: Egan, A. J. F., J. Biboy, I. van't Veer, E. Breukink & W. Vollmer, (2015) Activities and regulation of peptidoglycan synthases. Philosophical Transactions of the Royal Society of London B: Biological Sciences 370: 20150031].

5) In the subsection "L,D-transpeptidase activity of YcbB", a schematic of this reaction is shown, but no biochemical data is presented showing that the protein actually exhibits this enzymatic activity. What is the basis for this claim? Without the underlying data the reader does not know anything about the reaction or its rate. (Also, how active is YcbB when compared to similar L,D or D,D enzymes?)

6) In the subsection "Inactivation of YcbB by β-lactams", no biochemical data is presented to support this claim. Without this data the reader does not know anything about the rate or extent of inhibition.

---

## [Author Response]

*[…] Essential revisions:*

*A major limitation of the manuscript in its current form is that are some gaps in logic and omissions of critical data. For example, there is broad interest in the field in antibiotic resistance, and the central theme of this paper is the ampicillin-resistant generation of a complete cell wall. However, this reviewer is unfamiliar with that happening in nature, and as a result, most β-lactam related resistance is mediated by drug modifying enzymes and efflux – is that the case? For the sake of a balanced perspective, the authors should discuss any evidence that this bypass occurs, or if not, why not?*

In *E. coli*, as in Gram-negative bacteria, resistance of clinical isolates to β-lactams is mostly due to a combination of β-lactamases, efflux, and reduced penetration of the drug across the outer membrane. In Gram-positive bacteria, reduced affinity of the targets and β-lactamase production are the most frequent mechanisms.

The bypass resistance mechanism mediated by L,D-transpeptidases has not been reported to emerge in response to the selective pressure of antibiotics in natural conditions. However, L,D-transpeptidases have a major role in peptidoglycan synthesis in members of the genus *Mycobacterium* since 70% to 80% of the cross-links are of the 3→3 type (1, 2). Bypass of the D,D-transpeptidase activity of the PBPs has occurred in this phylum, although the modification of the peptidoglycan assembly pathway is ancient and most probably not linked to the selective pressure of β-lactams. The determinants of β-lactam resistance in the mycobacteria are complex. Resistance is not solely determined by the targets since the mycomembrane limits access of the drugs to the periplasm. In addition, production of a broad spectrum β-lactamase (BlaC) limits the efficacy of most, if not all, β-lactams, but BlaC is inactivated by the β-lactamase inhibitor clavulanate (3, 4). Regarding the targets, insertional inactivation of the gene encoding the L,D-transpeptidase Ldt_Mt2_ results in increased killing of *M. Tuberculosis* by the amoxicillin-clavulanate combination (5). However, L,D-transpeptidases are not the only target of β-lactams in *M. Tuberculosis* since the essential tetrapeptide is produced by D,D-carboxypeptidases that are inhibited by β-lactams (6). In conclusion, mycobacteria are the only naturally occurring bacteria in which L,D-transpeptidases have a role in resistance in natural isolates. Of note, we have shown that other members of the *Actinomycetales* produce a peptidoglycan containing similar amounts of 4→3 and 3→3 cross-links (7). In these bacteria, the L,D-transpeptidases may contribute to tolerance rather than resistance to glycopeptide antibiotics, which inhibit peptidoglycan synthesis by a mechanism unrelated to that of β-lactams (7).

In *E. faecium,* emergence of resistance to β-lactams by the bypass mechanism has been reported in laboratory conditions (8), as described in the current manuscript for *E. coli*. The large number of mutations observed in the mutants of *E. faecium* (9), combined to the slower growth associated with bypass of the PBPs (10), and the ease of selection of β-lactam resistant mutants by modification of low-affinity PBP5 (11) may limit the emergence of the bypass resistance mechanism in *E. faecium*.

In *E coli*, emergence of resistance was shown in the current manuscript to only require an increase in (p)ppGpp synthesis provided that YcbB is produced at an appropriate level. As indicated in the 3^rd^ paragraph of the Results section, attempts to select ampicillin-resistant derivatives of the strain M1_cured_ were negative (survivor frequency < 10^-9^). Thus, YcbB is not produced at a sufficient level for bypass of the PBPs in wild-type *E. coli* and the *ycbB* promoter could not be activated in a single selection step. This could be one of the key limitations for the emergence of the bypass mechanism in *E. coli*. In addition, the *E. coli* mutants described in our study displayed slowed growth, as reported in *E. faecium*, but the underlying mechanisms were different as increased production of (p)ppGpp is not involved in *E. faecium* (9, 12).

These observations indicate that the 3→3 mode of peptidoglycan cross-linking has a large fitness cost for exponential growth in laboratory conditions. It is therefore likely that a fitness cost is also associated with this mode of peptidoglycan cross-linking in natural conditions, although the possibility for acquisition of compensatory mutations has not been studied. Lessons from the past have indicated that certain resistance mechanisms considered to be highly unlikely have emerged after decades of antibiotic use (e.g. glycopeptides resistance due to modification of the structure of peptidoglycan precursors). Discussing the likelihood of the emergence of the bypass mechanism is highly speculative.

*In terms of leaps of logic in the experimentation that require further explanation:*

*1) As it currently reads (subsection “YcbB-mediated β-lactam resistance”, first paragraph), it appears that selection for both resistance to 500 μg/ml IPTG induction and resistance to 32 μg/ml ampicillin were performed simultaneously. Is this correct? If so, the rationale as well as rate of mutation should be better explained and documented, as this is central to the manuscript.*

The reviewer is correct in describing the conditions for selecting mutant M1. The survival rate was ca. 10^-9^ for the selection of mutants on media containing 500 µM of IPTG and 32 µg/ml of ampicillin. Induction of the gene encoding YcbB with 50 µM IPTG is sufficient for resistance, whereas higher levels of YcbB production are toxic. The LacI mutation reduced the expression of YcbB providing the right amount of YcbB.

The mutant M1 selected under those conditions harbored two mutations as determined by genetic analyses and confirmed by whole genome sequencing. The acquisition of two mutations in a single step might seem surprising if one considers the rate of mutations in *E. coli* (in the order of 1 X 10^-3^ per genome and per generation; (13)). However, several factors may increase the survival rate and/or the mutation frequency (see (14) for a discussion). It is also worth noting that one of the two mutations occurred on a gene carried by a multicopy plasmid (*lacI*) and that the mutation is likely to be dominant. We therefore think that the selection of survivors with two mutations did not involve any mutator phenotype, in agreement with the presence of only two mutations in the entire genome, as established by whole genome sequencing. We did not further comment on the basis for the frequency of survivors with two mutations in *E. coli* since this is beyond the scope of the current manuscript.

*2) Was an actual change in transcription in the lacI mutant seen or is this simply presumed? (And why is high expression of YcbB toxic (subsection “YcbB-mediated β-lactam resistance”, first paragraph)? The authors may not know, of course, but it would be useful to briefly speculate based on the authors understanding).*

The location of the amino acid substitution in LacI clearly indicates that binding of IPTG to the repressor is impaired thereby decreasing transcription of *ycbB* in response to the inducer. We do not think that this aspect requires experimental evidence in the context of the current manuscript. Regarding the toxic effect of YcbB, we have experimentally shown that induction of *ycbB* by IPTG in excess of 50 µM fully inhibits growth of the *E. coli* strains only producing the wild-type LacI repressor. Like high-molecular weight PBPs (D,D-transpeptidases), YcbB harbors a putative N-terminal membrane anchor. The toxicity generated by overproduction of YcbB may therefore be due to the insertion of the protein into the cytoplasmic membrane. Evidence for this mechanism of toxicity has been reported for overproduction of PBP2 in *E. coli* (15).

A sentence has been added to the revised version of the manuscript:

“The toxicity associated with high-level production of YcbB may be linked to the putative membrane anchor of the protein, as demonstrated for PBP2 in *E. coli* (15).” Modifications appear in the Results section.

*3) Was a change in ppGpp levels in the tRNA synthetase mutants that were essential for the phenotype actually measured? As there are broad effects of altering ppGpp pools, the lack of direct connection detracts from the perceived robustness of the results and the model proposed. There are also effects on the cell of mistranslation.*

We haven’t compared the ppGpp pools. An increase in the ppGpp pool in response to impaired aminoacylation of tRNAs has been documented in several publications (e.g. (16)). The fact that the mutants described in Table 4 harbor mutations in various aminoacyl-tRNA genes and that one of them harbors a mutation in the promoter region of the IleRS gene clearly indicates that the mutations affect the level of amino acylation. In addition, we have shown that expression of *relA’* has the same effect on expression of resistance. In this case, the pool of ppGpp has been determined (17). Together, these data are sufficient to conclude that an increase in the production of ppGpp results in resistance. We agree with the reviewer that direct determination of the ppGpp pool would provide an estimate of the intracellular concentration of the alarmone that is required for growth in the presence of ampicillin when YcbB is the only functional transpeptidase. We anticipate that the changes in the level of ppGpp that enable both resistance and growth might be rather subtle and think that this additional aspect, which is technically challenging, is not required to support the main conclusions of our study.

Regarding the effect on mistranslation, our experiment with RelA’ is performed in the absence of amino acid starvation. According to Sorensen et al. (18), the fidelity of translation is decreased in a *relA* strain, which is unable to down regulate mRNA synthesis with (p)ppGpp in response to amino acid starvation. Our mutants are expected to produce elevated ppGpp levels and are therefore not expected to display a decrease in the fidelity of translation.

*4) When the glycosyltransferase (GT) activity of PBP1B is blocked by mutation, its transpeptidase (TP) activity also decreases substantially or disappears (Egan et al., 2015). This means that the PBP1B-E^233^ mutant lacks both GT and TP activities. Thus, even if the GT enzymatic domain is "inactivated independently" (subsection “Identification of the glycosyltransferase partner of YcbB”, second paragraph), it is unclear how the authors can claim that only "the GT activity of PBP1b.… is required for.… β-lactam resistance" (second paragraph), or that this GT activity "is necessary and sufficient" (third paragraph). (Reference: Egan, A. J. F., J. Biboy, I. van't Veer, E. Breukink & W. Vollmer, (2015) Activities and regulation of peptidoglycan synthases. Philosophical Transactions of the Royal Society of London B: Biological Sciences 370: 20150031.)*

The reviewer is correct in stating that introduction of the E^233^M substitution in PBP1b is likely to impair the transpeptidase activity of PBP1b since in vitro analyses of PBP1a have shown that inactivation of the transglycosylase domain prevents formation of cross-links, although the transpeptidase domain remains functional with respect to acylation by β-lactams (19). Thus, we acknowledge the fact that the first version of the manuscript was erroneous in stating that we had selectively inactivated the transglycosylase and transpeptidase domains of PBP1b. We have therefore rephrased the entire paragraph (subsection “Identification of the glycosyltransferase partner of YcbB”, second paragraph) and cited the review article by Egan et al.(as suggested by the reviewer; (20)) and the references corresponding to the original publications on this topic (19, 21-23). We have also separately described our results obtained for production of PBP1b E^233^M in *E. coli* cells and mentioned the caveat for the potential loss of transpeptidase activity previously observed in vitro. However, we have maintained the conclusion that since this conclusion only requires to demonstrate that the transpeptidase domain is unessential, as experimentally shown in the preceding paragraph based on the S^510^A substitution.

*5) In the subsection "L,D-transpeptidase activity of YcbB", a schematic of this reaction is shown, but no biochemical data is presented showing that the protein actually exhibits this enzymatic activity. What is the basis for this claim? Without the underlying data the reader does not know anything about the reaction or its rate. (Also, how active is YcbB when compared to similar L,D or D,D enzymes?)*

The evidence is based on the identification of the reaction products (two dimers and a tripeptide) by mass spectrometry and determination of their structure by tandem mass spectrometry. The observed and calculated masses of the products were included in the first version of the manuscript. As requested by the reviewer, we have added a figure (Figure 5) showing the actual data. Regarding the rate of the reaction, the intensity of the peaks indicates that ca. 85% of the disaccharide-tetrapeptide substrate has been converted to products in 2 hours. Dimers were more abundant than the tripeptide produced by the L,D-carboxypeptidase activity of YcbB. More detailed quantitative assessment is not feasible since the dimer Tri-Tri can result from the formation of the dimer Tri-Tetra followed by cleavage of D-Ala^4^ or from the formation of a dimer using a disaccharide-tripeptide as the acyl acceptor. The kinetics are therefore too complex to provide rates for the L,D-carboxypeptidase and L,D-transpeptidase reactions. Similar analyses performed by Triboulet *et al.* (24) with the L,D-transpeptidase of *E. faecium* (Ldt_fm_) provided similar results with a seemingly lower L,D-transpeptidase to L,D-carboxypeptidase ratio (1.3) and higher overall activity. The differences are however not extensive (< 3 fold) and differences in the assays preclude accurate comparisons. In particular, the two substrates used for Ldt_fm_ were exclusively used either as an acyl donor or an acyl acceptor. A similar assay could not be developed for YcbB due to differences in the structure of the peptidoglycan from *E. faecium* and *E. coli*. Comparison with D,D-transpeptidases (PBPs) is not feasible since these enzymes are not active with disaccharide peptides, requiring coupling with transglycosylation and use of a lipid II intermediate as the substrate.

*6) In the subsection "Inactivation of YcbB by β-lactams", no biochemical data is presented to support this claim. Without this data the reader does not know anything about the rate or extent of inhibition.*

As requested by the reviewer, we have added a figure (Figure 7) showing the extent of acylation of YcbB by imipenem, meropenem and ceftriaxone.

References

1) Lavollay M, Arthur M, Fourgeaud M, Dubost L, Marie A, Veziris N, Blanot D, Gutmann L, & Mainardi JL (2008) The peptidoglycan of stationary-phase *Mycobacterium tuberculosis* predominantly contains cross-links generated by L,D-transpeptidation. J Bacteriol 190(12):4360-4366. DOI: JB.00239-08 [pii]10.1128/JB.00239-08.

2) Lavollay M, Fourgeaud M, Herrmann JL, Dubost L, Marie A, Gutmann L, Arthur M, & Mainardi JL (2011) The peptidoglycan of *Mycobacterium abscessus* is predominantly cross-linked by L,D-transpeptidases. J Bacteriol 193:778-782. DOI: JB.00606-10 [pii]10.1128/JB.00606-10.

3) Hugonnet JE & Blanchard JS (2007) Irreversible inhibition of the *Mycobacterium tuberculosis* β-lactamase by clavulanate. Biochemistry 46(43):11998-12004.

4) Hugonnet JE, Tremblay LW, Boshoff HI, Barry CEr, & Blanchard JS (2009) Meropenem-clavulanate is effective against extensively drug-resistant *Mycobacterium tuberculosis*. Science 323:1215-1218.

5) Gupta R, Lavollay M, Mainardi JL, Arthur M, Bishai WR, & Lamichhane G (2010) The *Mycobacterium tuberculosis* protein Ldt_(Mt2)_ is a nonclassical transpeptidase required for virulence and resistance to amoxicillin. Nat Med 16:466-469. DOI: nm.2120 [pii]10.1038/nm.2120.

6) Kumar P, Arora K, Lloyd JR, Lee IY, Nair V, Fischer E, Boshoff HI, & Barry CE, 3rd (2012) Meropenem inhibits D,D-carboxypeptidase activity in *Mycobacterium tuberculosis*. Mol Microbiol 86(2):367-381. DOI: 10.1111/j.1365-2958.2012.08199.x.

7) Hugonnet JE, Haddache N, Veckerle C, Dubost L, Marie A, Shikura N, Mainardi JL, Rice LB, & Arthur M (2014) Peptidoglycan cross-linking in glycopeptide resistant Actinomycetales. Antimicrob Agents Chemother 58: 1749-1756. DOI: AAC.02329-13 [pii] 10.1128/AAC.02329-13.

8) Mainardi JL, Legrand R, Arthur M, Schoot B, van Heijenoort J, & Gutmann L (2000) Novel mechanism of β-lactam resistance due to bypass of DD-transpeptidation in *Enterococcus faecium*. J Biol Chem 275(22):16490-16496. DOI: 10.1074/jbc.M909877199 M909877199 [pii].

9) Sacco E, Cortes M, Josseaume N, Bouchier C, Dubée V, Hugonnet JE, Mainardi JL, Rice LB, & Arthur M (2015) The mutation landscape of acquired cross-resistance to glycopeptide and β-lactam antibiotics in *Enterococcus faecium*. Antimicrob Agents Chemother. DOI: 10.1128/aac.00634-15.

10) Mainardi JL, Morel V, Fourgeaud M, Cremniter J, Blanot D, Legrand R, Frehel C, Arthur M, Van Heijenoort J, & Gutmann L (2002) Balance between two transpeptidation mechanisms determines the expression of β-lactam resistance in *Enterococcus faecium*. J Biol Chem 277(39):35801-35807. DOI: 10.1074/jbc.M204319200 M204319200 [pii].

11) Sifaoui F, Arthur M, Rice L, & Gutmann L (2001) Role of penicillin-binding protein 5 in expression of ampicillin resistance and peptidoglycan structure in *Enterococcus faecium*. Antimicrob Agents Chemother 45(9):2594-2597.

12) Sacco E, Cortes M, Josseaume N, Rice LB, Mainardi JL, & Arthur M (2014) Serine/threonine protein phosphatase-mediated control of the peptidoglycan cross-linking L,D-transpeptidase pathway in *Enterococcus faecium*. MBio 5(4):e01446-01414. DOI: 10.1128/mBio.01446-14.

13) Lee H, Popodi E, Tang H, & Foster PL (2012) Rate and molecular spectrum of spontaneous mutations in the bacterium *Escherichia coli* as determined by whole-genome sequencing. Proc Natl Acad Sci USA 109(41):2774-2283. DOI: 1210309109 [pii] 10.1073/pnas.1210309109.

14) Katz S & Hershberg R (2013) Elevated mutagenesis does not explain the increased frequency of antibiotic resistant mutants in starved aging colonies. PLoS Genet 9(11):e1003968. DOI: 10.1371/journal.pgen.1003968 PGENETICS-D-13-00671 [pii].

15) Legaree BA, Adams CB, & Clarke AJ (2007) Overproduction of penicillin-binding protein 2 and its inactive variants causes morphological changes and lysis in *Escherichia coli*. J Bacteriol 189(14):4975-4983. DOI: 10.1128/JB.00207-07.

16) Vinella D, D'Ari R, Jaffe A, & Bouloc P (1992) Penicillin binding protein 2 is dispensable in *Escherichia coli* when ppGpp synthesis is induced. EMBO J 11(4):1493-1501.

17) Schreiber G, Metzger S, Aizenman E, Roza S, Cashel M, & Glaser G (1991) Overexpression of the relA gene in *Escherichia coli*. J Biol Chem 266(6):3760-3767.

18) Sorensen MA, Jensen KF, & Pedersen S (1994) High concentrations of ppGpp decrease the RNA chain growth rate. Implications for protein synthesis and translational fidelity during amino acid starvation in *Escherichia coli*. J Mol Biol 236(2):441-454. DOI: 10.1006/jmbi.1994.1156.

19) Born P, Breukink E, & Vollmer W (2006) in vitro synthesis of cross-linked murein and its attachment to sacculi by PBP1A from *Escherichia coli*. J Biol Chem 281(37):26985-26993. DOI: 10.1074/jbc.M604083200.

20) Egan AJ, Biboy J, van't Veer I, Breukink E, & Vollmer W (2015) Activities and regulation of peptidoglycan synthases. Philos Trans R Soc Lond B Biol Sci 370(1679). DOI: 10.1098/rstb.2015.0031.

21) Terrak M, Ghosh TK, van Heijenoort J, Van Beeumen J, Lampilas M, Aszodi J, Ayala JA, Ghuysen JM, & Nguyen-Disteche M (1999) The catalytic, glycosyl transferase and acyl transferase modules of the cell wall peptidoglycan-polymerizing penicillin-binding protein 1b of *Escherichia coli*. Mol Microbiol 34(2):350-364.

22) Bertsche U, Breukink E, Kast T, & Vollmer W (2005) in vitro murein peptidoglycan synthesis by dimers of the bifunctional transglycosylase-transpeptidase PBP1B from *Escherichia coli*. J Biol Chem 280(45):38096-38101. DOI: 10.1074/jbc.M508646200.

23) den Blaauwen T, Aarsman M, & Nanninga N (1990) Interaction of monoclonal antibodies with the enzymatic domains of penicillin-binding protein 1b of *Escherichia coli*. J Bacteriol 172(1):63-70.

24) Triboulet S, Bougault CM, Laguri C, Hugonnet JE, Arthur M, & Simorre JP (2015) Acyl acceptor recognition by *Enterococcus faecium* L,D-transpeptidase Ldtfm. Mol Microbiol 98:90-100. DOI: 10.1111/mmi.13104.

25) Glauner B (1988) Separation and quantification of muropeptides with high-performance liquid chromatography. Anal Biochem 172(2):451-464.

26) Glauner B, Holtje JV, & Schwarz U (1988) The composition of the murein of *Escherichia coli*. J Biol Chem 263(21):10088-10095.

27) Blasco B, Pisabarro AG, & de Pedro MA (1988) Peptidoglycan biosynthesis in stationary-phase cells of *Escherichia coli*. J Bacteriol 170(11):5224-5228.

28) Pisabarro AG, de Pedro MA, & Vazquez D (1985) Structural modifications in the peptidoglycan of *Escherichia coli* associated with changes in the state of growth of the culture. J Bacteriol 161(1):238-242.

29) Vollmer W, Blanot D, & de Pedro MA (2008) Peptidoglycan structure and architecture. FEMS Microbiol Rev 32(2):149-167.